# Flexible ultrasound-induced retinal stimulating piezo-arrays for biomimetic visual prostheses

Laiming Jiang [1,2,3,5✉], Gengxi Lu [1,2,5], Yushun Zeng [1,2,5], Yizhe Sun [2], Haochen Kang[2], James Burford[1], Chen Gong[1,2], Mark S. Humayun[1,2,4], Yong Chen [3✉] & Qifa Zhou [1,2✉]

Electronic visual prostheses, or biomimetic eyes, have shown the feasibility of restoring functional vision in the blind through electrical pulses to initiate neural responses artificially. However, existing visual prostheses predominantly use wired connections or electromagnetic waves for powering and data telemetry, which raises safety concerns or couples inefficiently to miniaturized implant units. Here, we present a flexible ultrasound-induced retinal stimulating piezo-array that can offer an alternative wireless artificial retinal prosthesis approach for evoking visual percepts in blind individuals. The device integrates a two-dimensional piezo-array with 32-pixel stimulating electrodes in a flexible printed circuit board. Each piezo-element can be ultrasonically and individually activated, thus, spatially reconfigurable electronic patterns can be dynamically applied via programmable ultrasound beamlines. As a proof of concept, we demonstrate the ultrasound-induced pattern reconstruction in ex vivo murine retinal tissue, showing the potential of this approach to restore functional, life-enhancing vision in people living with blindness.

[1] Roski Eye Institute, Department of Ophthalmology, Keck School of Medicine, University of Southern California, Los Angeles, CA 90033, USA. [2] Department of Biomedical Engineering, Viterbi School of Engineering, University of Southern California, Los Angeles, CA 90089, USA. [3] Epstein Department of Industrial and Systems Engineering, Viterbi School of Engineering, University of Southern California, Los Angeles, CA 90089, USA. [4] Allen and Charlotte Ginsburg Institute for Biomedical Therapeutics, University of Southern California, Los Angeles, CA 90089, USA. [5] These authors contributed equally: Laiming Jiang, Gengxi Lu, Yushun Zeng. ✉email: laiming_jiang@foxmail.com; yongchen@usc.edu; qifazhou@usc.edu

The eyes are arguably the most important sensory organ for humans. Retinal degenerative diseases, such as retinitis pigmentosa and age-related macular degeneration, would cause loss of function of photoreceptors and result in significant visual deficits for the afflicted individuals[1–3]. There are few treatment options for patients with retinal photoreceptor degeneration[4]. Recently, the technological feasibility of replacing the photoreceptor function with electronic visual prostheses is being established by a growing body of research[5–7]. Electrically elicited phosphenes have been reliably demonstrated in early studies with retinal implant prototypes[8,9]. A typical retinal prosthesis system mainly involves an implantable stimulator and an external unit, with a wired connection or a wireless link between the two. A video camera in the external unit captures images and converts them to digital data. The command signals are appropriately encoded and transmitted to the implanted unit that receives the signals, recovers power and data from them, and sets the parameters of the stimulator. The stimulus patterns are then applied to the retina through the electrode array that directly interfaces the retinal surface[1]. In this context of retinal prostheses, a power and data transmission platform capable of delivering sufficient electricity for neuronal responses within security limits is required, while ensuring the equipment's portability, longevity, and cytocompatibility. However, existing visual prosthesis systems, such as Argus II System (Supplementary Fig. 1)[10,11], predominantly use a wired connection or electromagnetic (EM) waves for power supply and data telemetry[5,11–15], which raises safety concerns or couples inefficiently to miniaturized implanted units (in millimeter or submillimeter scale).

An alternative option for wirelessly driving implantable/wearable electronics is to employ programmable external ultrasound (US) sources, which would provide satisfactory controllability in terms of spatial resolution and electrical output parameters[16–18]. Recently, US-driven energy transfer has already established itself as an emerging technology for wireless power transfer and communication[19–23]. Compared with EM waves, US could allow wireless power and data transmission through acoustic waves with shorter wavelengths (e.g., millimeter and submillimeter at 1–10-MHz US frequency)[21,24]. For example, 3.3-MHz US waves possess a wavelength of 455 μm, while the wavelength of 3.3-GHz EM waves is 15 mm. The shorter wavelength enables efficient coupling with tiny electronics, especially miniaturized array receivers, such as a biomimetic retinal stimulating array. US also allows a higher U.S. Food and Drug Administration (FDA) regulatory limit on power flux density compared with EM waves ($720\,\mathrm{mW\,cm^{-2}}$ versus $10\,\mathrm{mW\,cm^{-2}}$)[25]. In addition, US possesses higher security in cybercommunications, with fewer potential cybersecurity vulnerabilities that are vulnerable to hacking attacks leading to system crashes[26–28]. Consequently, ultrasonic efficient coupling and high power limit ensure highly accurate control of stimulus charge that is vital for targeting repeatable neuronal populations and maintaining safety throughout chronic use (Supplementary Table S1). Our previous work presented a millimeter-level ultrasonic energy device for a potential retinal electrical stimulation scheme that used US-induced wireless power delivery technology to convert acoustic energy into electrical energy, yielding current signals higher than the thresholds of retinal electrical stimulation[29]. Nevertheless, the simple architecture with a single piezo-element limited the visual projection of complex patterns, significantly restricting its broad application.

Here, we report a sophisticated flexible US-induced retinal stimulating piezo-array (F-URSP) that may provide prosthetic vision to people with acquired blindness. The device integrates a two-dimensional (2D) piezo-array, rectifiers, and a stimulating electrode array with 32 pixels in a flexible printed circuit board (PCB). The US fields with the distribution of acoustic power in space can be appropriately programmed and wirelessly transmitted through an external US transducer, such as a 2D array. The emitted ultrasonic power will be received by the piezo-array and then be converted to electricity to initiate a neural response in cell populations of the retina. Spatial complex stimulus patterns can be applied to the retina through the stimulating electrode array, which is composed of separate electrodes that interfaces at many locations on the retinal surface. Furthermore, by optimizing the architecture, the high acoustic-electric conversion efficiency was obtained in each 1-3 piezo-element, and the electrical threshold ($\approx$1–50 nC)[1] for evoking a retina response can be induced under a low and safe ultrasonic intensity. As a proof of principle, we demonstrated the continuous acoustic field-induced pattern reconstruction and electrical stimulation response of the ex vivo murine retinal tissue by using the F-URSP. To examine the device's cytocompatibility, we further cultured prostate cancer cell (PC-3) lines on the substrates of the F-URSP. The PC-3 cell lines exhibit a viability (>96.6%) comparable to the petri-dish control groups, indicating that the F-URSP does not exhibit cytotoxicity. These studies establish the foundation for US-induced wireless visual prostheses, which provide a method in creating an artificial vision for people living with blindness.

## Results

### Device design, working principle, and structural optimization.
Figure 1a schematically illustrates the design and working principle of the F-URSP. The device integrates a 1-3 piezo-composite array for acoustic-to-electric conversion, rectifiers for alternating current-direct current (AC-DC) converters, and a stimulating electrode array with 32 pixels for direct contact with the retina in a soft, flexible PCB, which consists of patterned layers of gold on a film of polyimide served as a substrate for these components. The F-URSP design refers to the size of the human eyeball (Fig. 1b and Supplementary Fig. 2) and is targeted to implant application as a retinal prosthesis for restoring the image-acquiring properties through the US-induced wireless transfer technology. The ultrasonic beamlines are programmed and transmitted through an external transmitter, such as wearable glasses integrated 2D array transducers, which are portable and require no physical connection with the implanted devices. Degassed US gel is applied as the coupling medium between the US transmitter and the receiving piezo-array for effective ultrasonic transmission and coupling. When hit by the ultrasonic beamlines, the piezo-array will vibrate and generate electric charges in response to applied US pressure. The generated alternating current (AC) signal will be further converted into a direct current (DC) signal (monophasic pulses are often used in animal experiments for studying neuronal responses to the stimulations) by the integrated rectifiers for stimulating localized neural populations in the retina, which conduct action potentials through the optic nerve to the central visual pathway to evoke phosphenes or artificial visual percepts (fundamentals of retinal electrical stimulation is shown in Supplementary Note 1). In other words, information can be transmitted from the retina to higher visual areas of the blind if their retinal ganglion cells and optic nerve are intact and functional[3]. In addition, there is an orderly mapping between each location in the visual scene and each location in the brain[4]. Each piezo-channel (CH1-CH32) in the F-URSP can be individually activated and controlled. Consequently, the primary competitive advantage of the presented F-URSP is the capability of selectively stimulating neural populations at distinct locations to realize phosphenes with arbitrary patterns by spatially programming US fields. In addition, the brightness and duration of a phosphene could also be manipulated by regulating the US

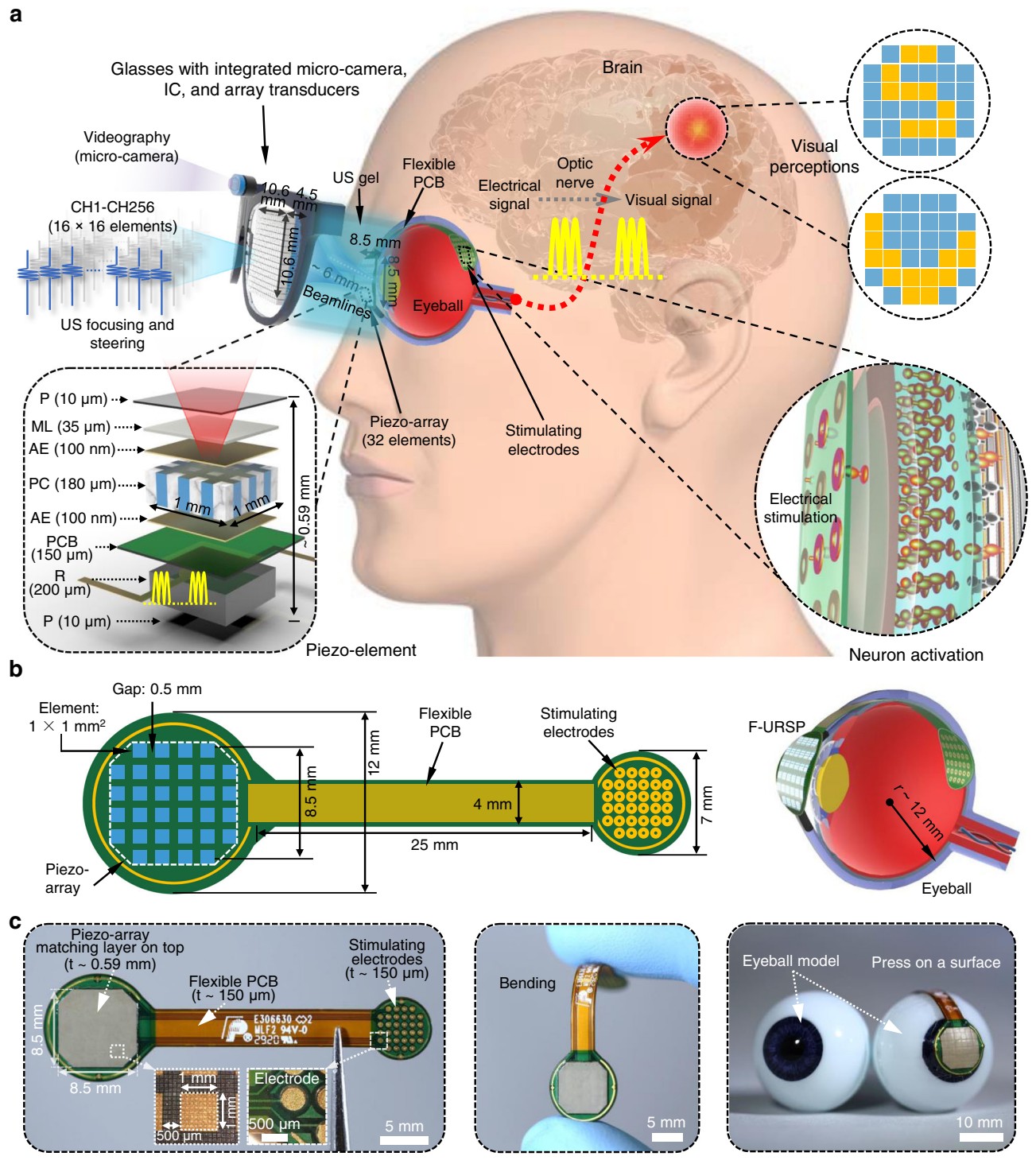

**Fig. 1 Design structure and working principle of the F-URSP. a** Schematic diagram showing the flexible ultrasonic device, with key components and aspect ratios labeled. The implantable F-URSP consists of a high-performance piezo-composite array, rectifiers, and an electrode array integrated into a flexible PCB. Each piezo-channel (CH1-CH32) in the array works individually and can convert the transmitted US wave into electricity which is rectified and then used for electrical stimulation of retinal neurons through the integrated electrodes. The correspondingly induced action potentials would be conducted to the central visual pathway via the optic nerve to produce visual perception. Therefore, the F-URSP architecture leverages the advantage of the programmable US beam to realize the visual projection of complex patterns wirelessly. A micro-camera for videography and an integrated chip (IC) for optical picture-to-program processing will be designed and integrated into the glasses in the future to excite the array. P, Parylene-C/coating (10 μm), protection for PZT; ML, matching layer (35 μm), enhancement of acoustic transmittance; PC, piezo-composite (180 μm), acoustic-electric conversion element; PCB, printed circuit board (150 μm), flexible circuit carrier; R, rectifier (200 μm), AC-DC conversion; AE, Au/Cr electrode (100 nm), electrodes of the piezo-element. **b** Schematic diagram showing structure and dimensions of the F-URSP. The design refers to the size of the human eyeball (radius ~ 12 mm) for future implant applications. **c** Photographs showing the as-fabricated final F-URSP with soft configuration (t, thickness).

waveforms (to induce different electrical waveforms)[8]. The as-fabricated final device is shown in Fig. 1c, highlighting its light-weight electronic design and soft, flexible mechanical properties, with the capability of achieving conformal contact with non-planar surfaces of real components, for example, an eyeball model of equal scale.

The performance of the integrated F-URSP requires reliable electricity generation from the transmitted US waves, that is, high acoustic-to-electric conversion efficiency. Therefore, lead magnesium niobate-lead titanate (PMN-PT) single crystal, which is known for its high electromechanical coupling coefficient ($k_t \sim 60\%$), high piezoelectric coefficient ($d_{33} \sim 1200$ pC N$^{-1}$), and low dielectric loss ($\tan\delta \leq 0.6\%$)[30,31], was selected as the piezoelectric material. The 1–3 composite structure that consists of periodic piezo-micropillars embedded in a passive resin matrix was further designed and fabricated via a dicing-and-filling method to optimize the device performance (Fig. 2a)[32,33]. Its acoustic impedance and electromechanical coupling performance can be tailored through varying the volume fraction and aspect ratio of piezo-micropillars, especially the improved electromechanical coupling in the thickness direction (Supplementary Fig. 3)[34]. The acoustic and electrical characteristics (e.g., acoustic impedance $Z_a$, permittivity $\varepsilon$, electromechanical coupling coefficient $k$, and voltage factor $g$) of the 1–3 composite can be predicted theoretically based on the series and parallel models proposed by Chan and Unsworth[35]. Several guidelines should be followed when designing the 1–3 piezo-composite to improve the acoustoelectric conversion efficiency when designing the 1–3 piezo-composites. First, the ratio of height to width of PMN-PT pillars needs to be >2. In this way, the half-wave transverse vibration of the piezo-pillars is much higher than the fundamental thickness one, so the thickness vibration coupling will dominate. Second, a resonant frequency twice higher than the fundamental thickness mode should be required for the stopband edge[36]. Accordingly, the kerf between the piezo-pillars should be less than $c_s/2f$, wherein $c_s$ and $f$ are the shear wave velocity of the filler and the thickness resonant frequency of the piezo-pillars, respectively.

According to the above theoretical model, a 1–3 PMN-PT/epoxy composite with miniature piezo-pillars (width: 90 μm, height: 180 μm) and kerfs (20 μm) was designed and manufactured (Fig. 2b). A matching layer (acoustic impedance ~6 MRayl) composed of epoxy resin containing silver powder was further deposited onto the piezo-composite to compensate for acoustic impedance mismatch between the composite (~10.6 MRayl) and US gel (~1.5 MRayl) and thus improved US energy transmission (Supplementary Fig. 4 and details in Supplementary Note 2)[37]. The acoustic stacks were divided into sub-blocks with an element footprint of 1 mm × 1 mm and a spacing of 500 μm (Fig. 2b). The whole piezo-layer with 32 elements and rectifiers was then bonded to the pre-prepared flexible PCB (~150 μm in thickness) using conductive silver paste. Finally, a 10-μm Parylene-C layer that electrically and mechanically protects the electronics and provides a barrier to biofluids was uniformly deposited over the device (except for the stimulating electrodes)[38]. For each piezo-element, an equivalent RLC circuit describes the electrical effect of its mechanical motion (Fig. 2c, inset)[39,40]. The electrical impedance and phase angle spectra of a representative element were measured (Fig. 2c), from which a resonant frequency located at 5.43 MHz was obtained. The corresponding electromechanical coupling coefficient $k_t$ and voltage coefficient $g_{33}$ were calculated (Fig. 2d), which were substantially enhanced (e.g., $g_{33} \sim 40.3 \times 10^{-3}$ V m N$^{-1}$, $k_t \sim 0.84$) through the 1-3 structure design, ensuring high acoustic-to-electric conversion efficiency of the device. In addition, the simulation results show that the ultrasonic receiving sensitivity (Fig. 2e) and the average piezo-

potentials (Fig. 2f, g) induced in the 1–3 composite are significantly higher than those in bulk PMN-PT crystal due to the suppressed shear vibration mode and the increased coupling coefficient, which dominates in the fundamental length longitudinal 33-mode with higher efficiency. The key electrical and acoustic parameters of the PMN-PT crystal and the prepared composite are listed in Supplementary Table 2.

**Performance validation**. The wireless energy transfer strategy of our F-URSP uses US to carry the available energy. Three primary parts, namely ultrasonic excitation, transmission, and reception, are involved in the process (Fig. 3a). First, a 3.3-MHz focused transducer (see Supplementary Note 3 for the frequency selection) that is composed of a lead zirconate titanate (PZT) piezo-ceramic plate sputtered on both sides with electrodes was designed and fabricated for ultrasonic excitation (Supplementary Fig. 5). The Field II simulation of the acoustic field was carried out for this transducer, as shown in Fig. 3b and Supplementary Fig. 6 (the detailed simulation method is shown in Supplementary Note 4). Benefiting from the focused piezo-architecture, the acoustic power is focused into a small region through the confined US beam, which not only enhances the magnitude of the acoustic excitation but also effectively improves the lateral resolution of the US beam[41]. The −6 dB lateral resolution is ~390 μm near the focus point (detailed calculation is shown in Supplementary Note 5), which is far below one element size, thus ensuring the excellent shape recognizability of the F-URSP. For the received US, theoretically, the pressure-induced output voltage $\bar{V}$ of a piezo-harvester can be written as[42]

$$\bar{V} = -\frac{Pk^2}{e_{33}(1+k^2)} \frac{Z_L}{\xi \cot(\xi h)(Z_0 + Z_L)} \tag{1}$$

wherein $P$, $k$, $e_{33}$, $\xi$, $Z_L$, $Z_0$, and $h$ are the applied pressure, electromechanical coupling coefficient, piezoelectric constants, wavenumber, load resistor, electric impedance, and height of the piezo-layer, respectively. The US-induced voltage is highly dependent on the applied pressure and the electromechanical coupling coefficient. 1-3 piezo-composite was therefore designed and fabricated to improve the output performance.

To experimentally prove the feasibility and accuracy of US-induced energy transmission, we measured the output voltage of the F-URSP (Supplementary Fig. 7) and compared the trigger signal and unrectified output signal waveforms, as shown in Fig. 3c. Under the excitation of a 3.3-MHz US pulse with a trigger voltage of 10 Vpp and repeat cycles of 200, the induced signal shows the consistent repeat period (~0.3 μs), the same pulse length (~60.6 μs), and a maximum output voltage of 1.5 Vpp. In addition, a typical time delay (~16 μs) and a reflected phase were detected in the output waveform as a result of US travel and reflection between the transmitter and receiver interfaces, further illustrating that the output signal is induced by the transmitted US.

Figure 3d shows the magnitudes of rectified output open-circuit (OC) voltage for a representative element measured as a function of the trigger voltage from 0 to 100 Vpp to evaluate the output voltage efficiency. The output voltage magnitudes increased in response to increasing trigger settings. An almost linear dependence was detected (Fig. 3e). For example, the induced unrectified output voltage reached ~10.6 Vpp under a trigger of 100 Vpp (Supplementary Fig. 8). The corresponding rectified voltage reached ~4.3 V, which is higher than the stimulus parameter of a electrode voltage of ~0.8 V used by a commercially available visual prosthesis[43]. The output measurements enable the extraction of the voltage efficiency (output voltage/input voltage, Vpp/Vpp) of each element in the piezo-array. All 32 piezo-

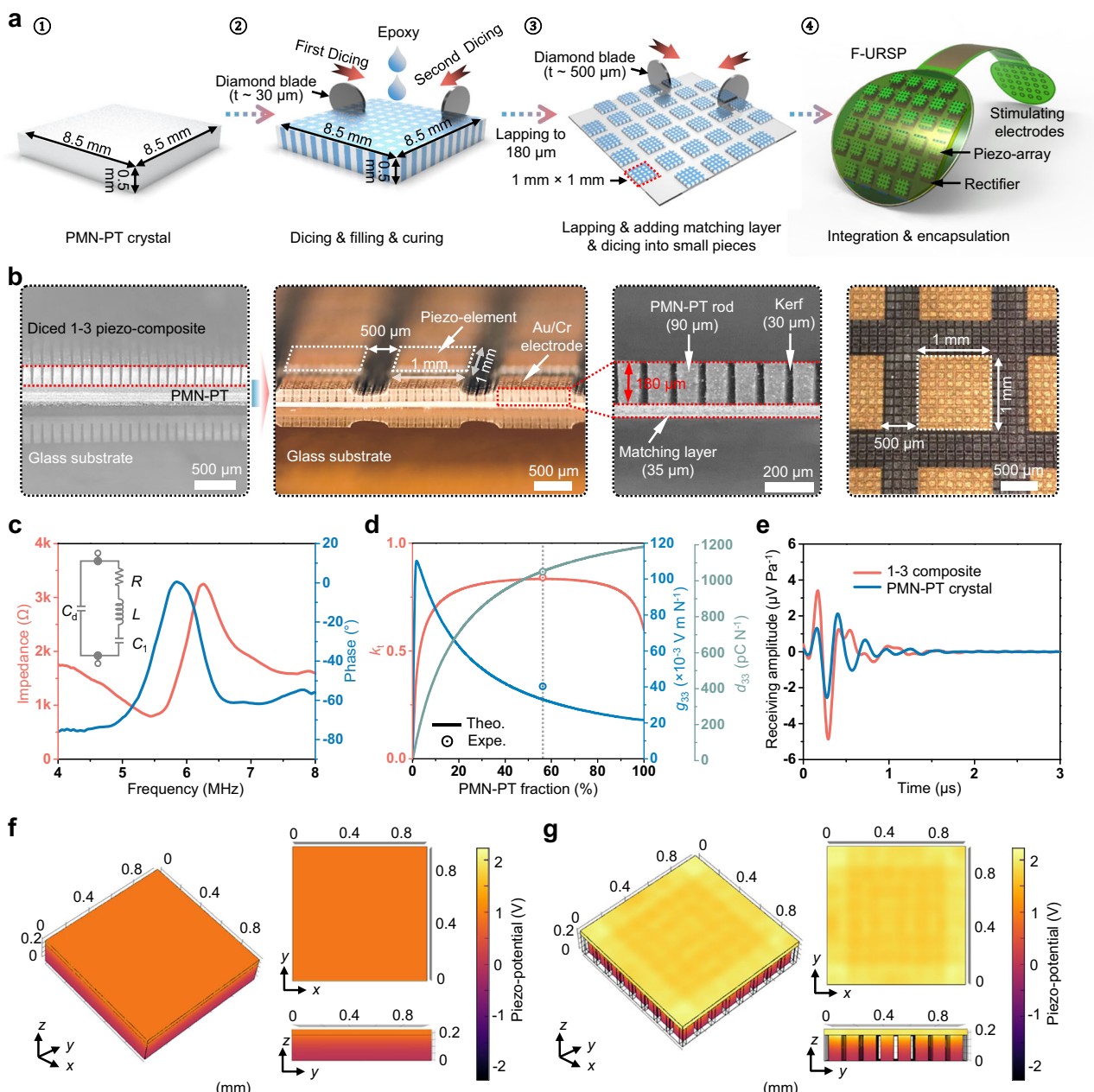

**Fig. 2 Structural optimization and simulation of the F-URSP. a** Schematic diagram showing the preparation process of the PMN-PT 1-3 composite for the harvesting array using a dicing-and-filling technology. **b** Photographs of the diced PMN-PT 1-3 composite and the prepared harvesting piezo-elements. The composite consists of 90-μm-wide PMN-PT rods and 30-μm-wide kerfs. Each element possesses a 1 mm × 1 mm element footprint with a spacing of 0.5 mm. **c** Impedance and phase angle spectra of a 1-3 composite piezo-element. Inset: equivalent $RLC$ circuit diagram of the piezo-element. **d** Theoretically calculated and experimentally measured electromechanical coupling coefficient ($k_t$), piezoelectric voltage coefficient ($g_{33}$), and piezoelectric constant ($d_{33}$) of the prepared 1-3 composite. The lines represent the theoretically calculated (Theo.) values and the symbols represent the experimentally measured (Expe.) values. **e** Simulated receiving sensitivity of the bulk PMN-PT piezo-element (~2.56 μV Pa$^{-1}$) and 1-3 composite element (~4.87 μV Pa$^{-1}$). **f**, **g** Simulated piezo-potentials inside a bulk PMN-PT piezo-element (**f**) and a 1-3 composite element (**g**) induced by the same acoustic pressure (0.5 MPa). Source data are provided as a Source data file.

channels are functional. The mean of voltage efficiency value is ~11.3%, with a standard deviation (SD) of 0.75% (Supplementary Fig. 9). The performance arising from the optimization of the 1-3 composite structure ensures high sensitivity in further electrical stimulation. Another important metric for assessing the performance of the piezo-array is the power density, which indicates the amount of energy that the device can provide. A series of voltage measurements were therefore performed when the device was connected to various external loads (Supplementary Fig. 10). As

the resistance increases, the output voltage increases progressively and then saturates at higher external loads. The corresponding instantaneous power density was calculated (Fig. 3f). For example, an instantaneous power density of up to 7.2 mW cm$^{-2}$ can be achieved for the piezo-array under a trigger voltage of 50 Vpp and an external load of 220 Ω. The spatial peak temporal average intensity ($I_{SPTA}$) at the face of receiving array alongside the trigger voltage is shown in Supplementary Fig. 11 and the US-induced outputs can also be flexibly adjusted to reach a power density up

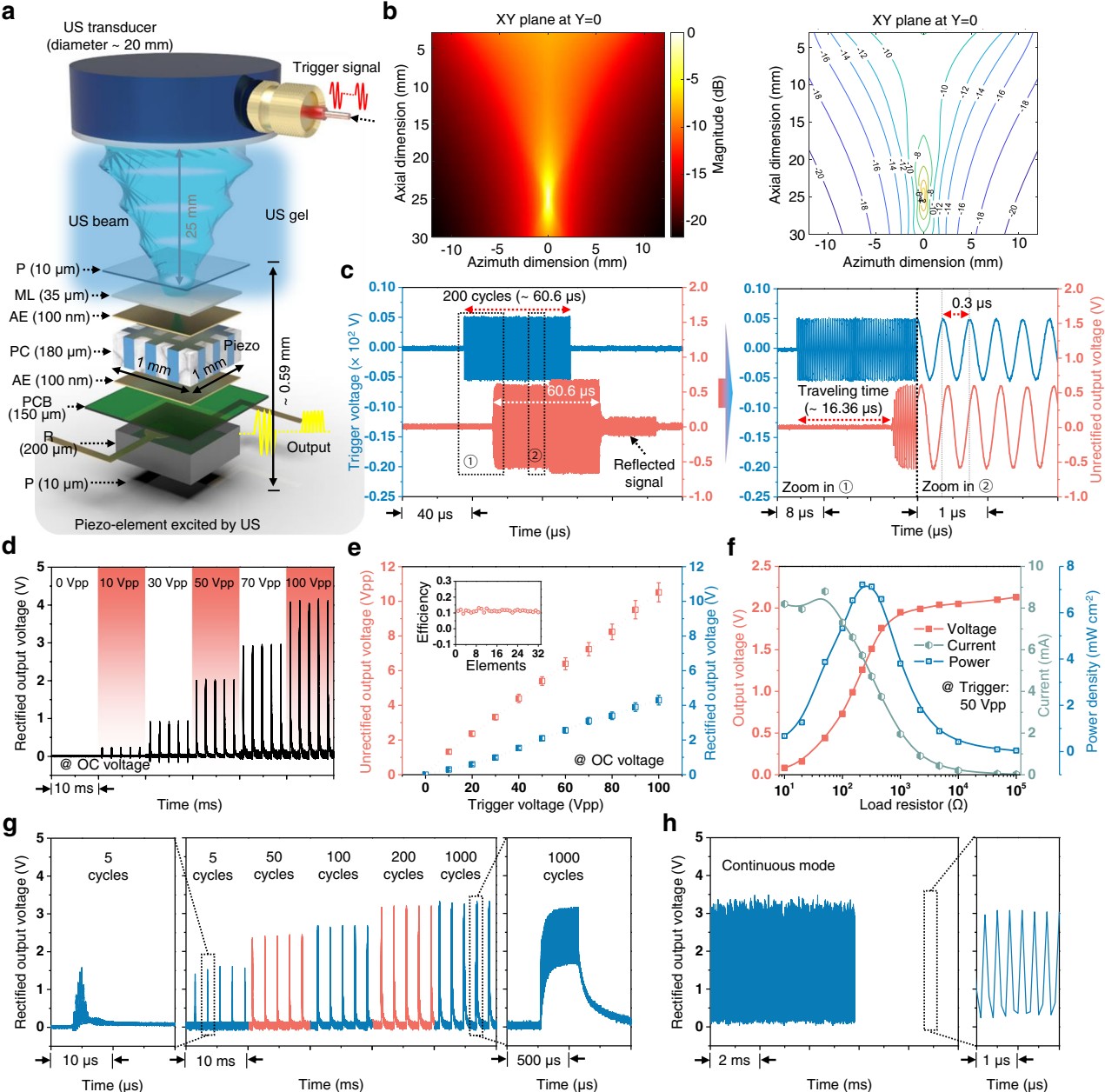

**Fig. 3 Simulation and characterization of US-induced electrical outputs. a** Schematics of an US-induced measurement setup, with a 3.3-MHz focused probe as US transmitter and with key components and aspect ratios labeled. **b** Simulated US field emitted by the focused probe using a Field II software package, showing a spatially confined US beam. The color bar indicates the magnitude distribution of acoustic pressure. **c** Comparison between the trigger signal used to excite the transducer and the unrectified output signal from the F-URSP. **d** Rectified output voltage magnitudes generated from the F-URSP using different trigger voltages. **e** Unrectified and rectified output voltages against the trigger voltages. *N* = 32 measurements of independent piezoelectric element. Data were presented as mean ± SD. Inset: The voltage efficiency variation of the 32 piezoelectric elements. **f** Output voltage, current, and corresponding power density of a piezo-element under a trigger voltage of 50 Vpp and various external loads. **g** Output voltages under the excitation of trigger signals with different repeat cycles (5–1000). **h** Output voltage under continuous mode. The results demonstrate that the US-induced energy can be flexibly adjusted by varying the duty cycles. Source data are provided as a Source data file.

to 22.6 mW cm$^{-2}$ through trigger parameters (Supplementary Fig. 12). To further demonstrate the output capability of the device, the US-induced energy was directly stored in a capacitor and then used to power a commercial LED (Supplementary Fig. 13). A 100 μF capacitor was charged by a piezo-element to 0.59 V in 50 s and 1.34 V in 200 s, at an average charging rate of 670–1180 nC s$^{-1}$. For comparison, the results from previous animal studies and chronic human implants demonstrate that a stimulus charge of ~1–50 nC will effectively evoke a visual response in the retina (Supplementary Table S3)[1,44,45]. In

addition, the trigger interval and pulse length of the induced signal can also be flexibly adjusted and programmable by varying the duty cycles for an on-demand control platform for retinal stimulation (Fig. 3g, h and Supplementary Figs. S14–S15).

**Pattern recognition induced by US field.** The prepared F-URSP, which functions as a biomimetic eye, relies on a large-scale piezo-array to realize the function of perceiving images (Fig. 4a). The device can achieve a conformal curvature with an eyeball model

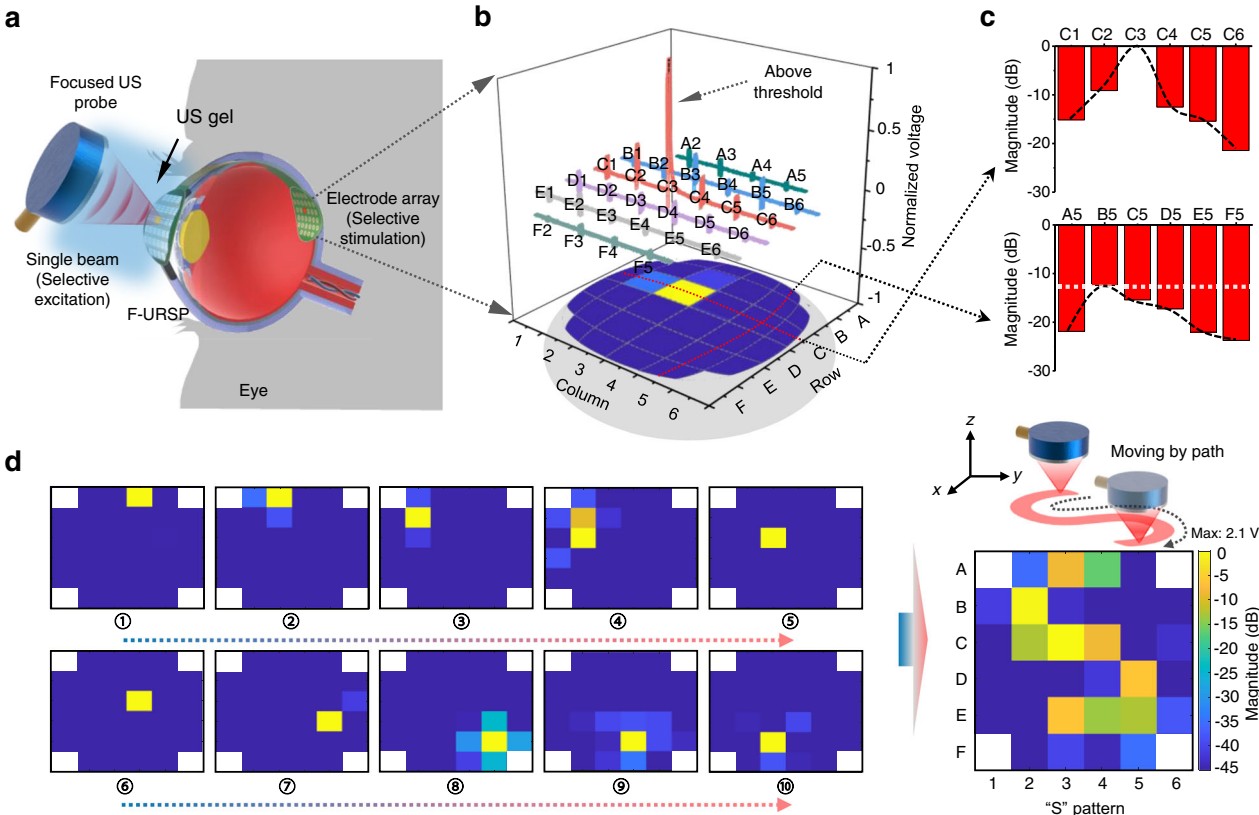

**Fig. 4 Recognition function of the F-URSP induced by a focused transducer. a** Schematic diagram showing one working model of the F-URSP, in which a focused US probe was used for selective excitation of the piezo-array. **b** Under the excitation of a single focused probe, the output voltage magnitude of each element in the array. **c** Line profile data showing the signal-to-noise ratio of the output voltage magnitude for two typical positions-one on a row with excitation elements (horizontal line) and one off the excitation (vertical line)-showing unambiguous differences. **d** Voltage magnitude distribution imaging result by the F-URSP. The image was reconstructed in a time domain with the single probe moving following a programmable S-shaped path. The maximum output voltage is 2.1 V. The color bar indicates the magnitude distribution ($20\log\frac{V_O}{V_{max}}$). Source data are provided as a Source data file.

of equal scale via a pressing process, demonstrating potential application in wearable/implantable electronic visual prostheses. The good uniformity of all the elements was verified by making statistics on the US-induced output voltages (Supplementary Fig. 9). First, a 3.3-MHz focused transducer that can emit a confined US beam with a lateral resolution of ~390 μm was employed as an acoustic source for obtaining a small spot with high power density distribution (Supplementary Fig. 6). As shown in Fig. 4b, the piezo-element that is perpendicularly incident by US beam unambiguously exhibits a high output voltage with a high signal-to-noise ratio (SNR) of ~32.6 dB. The other elements that are not incident perpendicular to the US beam show significant attenuation in magnitude due to reduced acoustic intensity. For the adjacent elements, a mean attenuation is ~−10.8 dB. In addition, Fig. 4c depicts the magnitude statistics of 11 elements, as indicated for two lines in Fig. 4b—one on the vertical incidence point (row C) and another off the vertical incidence point (column 5). The magnitudes of these elements for ultrasonic normal incidence or not are distributed in a broad range, showing the gigantic effect of the US intensity on piezo-potential. Subsequently, to demonstrate the capability of pattern reconstruction, the F-URSP was laminated on hemispherical support and the focused probe was applied to create an active area by moving in a programmed path (Supplementary Fig. 16). The voltage magnitude of each pixel was collected and analyzed through the fast-testing pixel by pixel. The pixels on the acoustic path can generate a higher potential (~2.1 V) above the voltage threshold of ~0.8 V for retinal excitation[43], while the other pixels

outside the focal spot receive less acoustic excitation and, therefore, produce lower electrical outputs with voltages below 0.8 V. As a result, a clear S-shaped pattern can be identified from the reconstructed mapping result that was US-induced in a time domain with the transducer moving following a programmable path (Fig. 4d).

To further demonstrate the shape reconstruction capability of the device, a 2D US array transducer with a 3.5-MHz center frequency and 16 × 16 elements was used to emit programmable acoustic fields for pattern recognition (Fig. 5a, b). Compared with a single probe in the focused or plane architecture, the multi-element 2D array transducer exhibits the scalability and programmability to allow steering of multiple ultrasonic beam-lines (Supplementary Fig. 17) in the viewing direction to generate arbitrary patterns by modifying the transmitting amplitude and phase of each element (Fig. 5c)[46–48]. Acoustic field simulations were performed via Field II to intuitively demonstrate ultrasonic field intensity distribution of complicated patterns at different depths, as shown in Fig. 5d, e and Supplementary Fig. 18. 3D acoustic pressure field configurations of letter-like patterns (e.g., H and V) exhibit high-intensity distributions in desired pattern areas. The simulated results fully prove that the programmable acoustic beams could be emitted based on the multi-element 2D array transducer. In addition, the transducer is not only used for the emission of programmable acoustic fields, but also for achieving US imaging to evaluate the alignment of the transmitter and receiver, as shown in Supplementary Fig. S19. Depending on the results of US imaging, the relative position of the transmitter

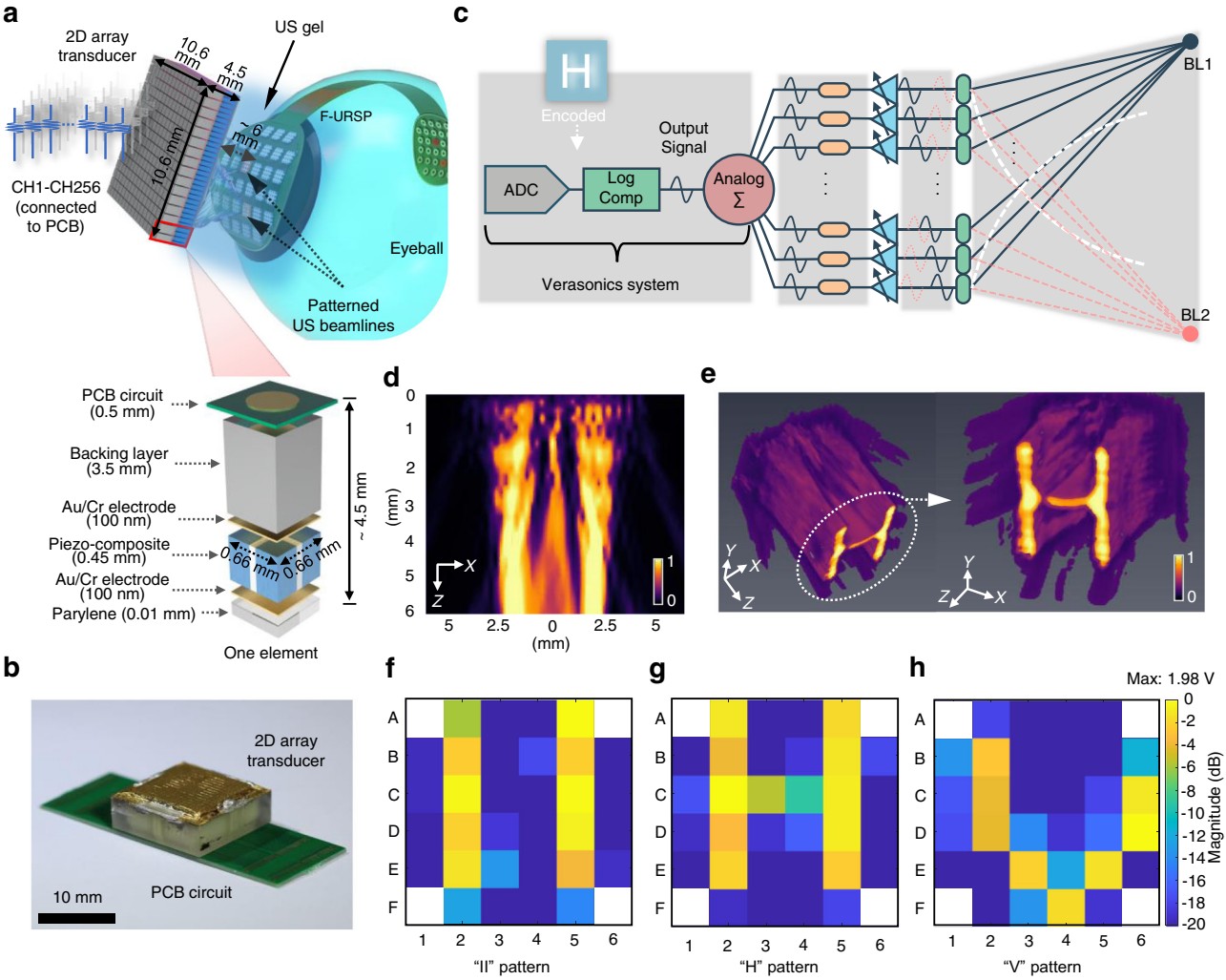

**Fig. 5 Recognition function induced by programmable acoustic field of a 2D array transducer. a** Schematic diagram showing one working model of the F-URSP, in which a 2D-array US probe was used to excite programmable acoustic fields, with key components and aspect ratios labeled. **b** Photographs of a 2D array transducer. **c** Schematic diagram showing the US beam steering and focusing with a transducer array by modifying the transmitting amplitude and phase of each element. **d** Simulated pressure magnitude distribution at the X–Z plane of a "H" pattern. **e** Simulated 3D acoustic intensity distribution of a "H" pattern emitted by a 2D array transducer. The color bar (**d**, **e**) indicates the normalized acoustic pressure. The maximum value is 0.5 MPa. **f–h** Voltage magnitude distribution imaging results, which were reconstructed via programmable II-shaped (**f**), H-shaped (**g**), and V-shaped (**h**) acoustic field patterns generated by a 2D US array transducer. The maximum output voltage is 1.98 V. The color bar indicates the magnitude distribution ($20\log\frac{V_O}{V_{max}}$). Source data are provided as a Source data file.

that is fixed on the 3D adjustable mount can be flexibly adjusted to align it with the receiver prior to the emission of acoustic fields. Then, a 3.5-MHz tone sine-wave burst with 10% duty cycle at 50 Vpp was employed by a Verasonics system to experimentally drive the 2D array transducer per trigger period. The focal distance and steering angle were electronically adjusted in real-time using a customized script. Combining the projection planes of all the beamlines can quickly produce programmable images to F-URSP[41]. As shown in Fig. 5f–h, letters of "II"-, "H"-, and "V"-shaped patterns were identified from the mapping results. The frame rate can reach to 1 kHz in burst. To demonstrate reliability in image reconstruction, the pattern recognition experiment was repeated and the mean squared error (MSE) was calculated for each pixel (Supplementary Fig. S20). The MSE values below 0.1. The error mainly comes from the uniformity of the emitted acoustic field. These can be further improved in the future by optimizing the 2D array transducer. In addition, although spillovers to nearby pixels are observed because the US beam is usually accompanied by sidelobes, we can control the amplitude

of US input, ensuring the main focusing region can activate neurons while sidelobes are weaker than the threshold, to avoid out-of-focus activation.

**Living retina stimulation and cytocompatibility.** The feasibility of US-induced retinal electrical stimulation was evaluated by using fluorescent imaging to investigate the $Ca^{2+}$ transients that are generally employed to acquire the activities of neuronal populations (Supplementary Note 6). The living retina dissected from the Ai95(RCL-GCaMP6f)-D transgenic mice was employed as a model system (Fig. 6a). An acoustic "V"-shaped pattern was provided by using the 2D US array transmitter (Fig. 6b). An electrical stimulus parameter using tone sine-wave bursts with 1000 cycles at 50 Vpp was applied for ultrasonic excitation of the 2D array (detailed experiment process in the "Methods" section). The confocal image of a collection area of the retina at stimulated levels was shown in Fig. 6c. The correspondingly observed fluorescence-enhanced region on the retina matches the expected excitation position (a "V"-shaped pattern), albeit with slight distortion due to the

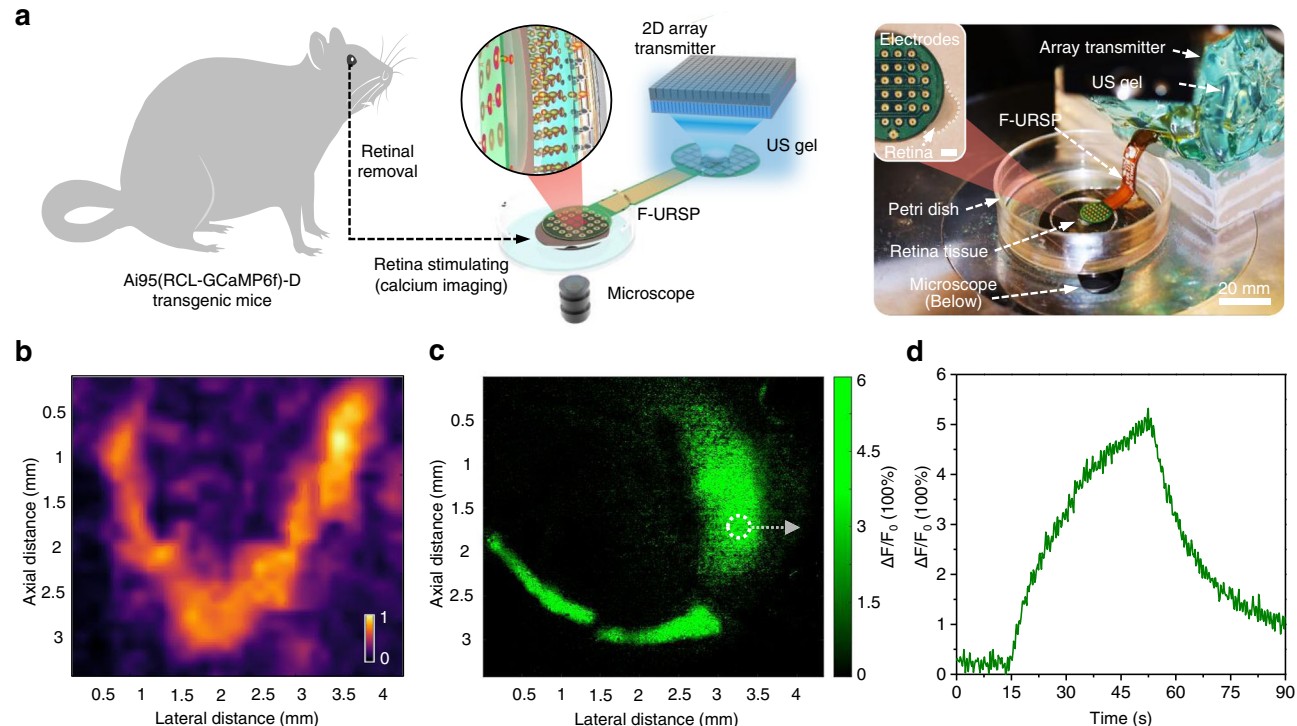

**Fig. 6 Living retina stimulation of F-URSP. a** Schematic diagram and photograph showing the ex vivo murine retina tissue and experimental setup for the retina stimulation using the F-URSP. The retina tissue was removed from the eye of an anesthetized mouse and stimulated using F-URSP. The inset shows the living retina placed under the stimulating electrodes of the F-URSP. Scale bar: 1 mm. **b** Measured acoustic pressure distribution of a "V"-shaped pattern emitted by a 2D array transducer through a hydrophone probe. The color bar indicates the normalized acoustic pressure. The maximum value is 0.45 MPa. **c** Calcium imaging of the living retina at stimulated levels. **d** Time course of calcium transients during stimulation. Source data are provided as a Source data file.

number of array elements and resolution limitations. Fluorescent intensity values were acquired from representative somas, which demonstrated an increase ($\Delta F/F_0 \sim 5\%$) in mean fluorescence intensity elicited by US-induced electrical pulses (Fig. 6d). It is a signal for the actual neural population activity and validates the feasibility of the US-induced scheme. US-induced electrical stimulation causes the same neuron responses as conventional electrical stimulation methods. For each 0.3 ms stimulation, neuron spikes last for around 10 ms; for the evoked calcium signals, it can generally last for 30 s[49]. Supplementary Fig. 21 schematically illustrates the fundamentals of electrical stimulation to the retina. The electrode array was placed on the retina tissue to form an electrochemical interface with saline. The current injected by the US-induced electrode travels through the tissue to the return electrode. The electric currents delivered to the extracellular area redistribute the charge across the cell membrane of the retinal neurons. The action potential is therefore initiated as soon as the membrane depolarization exceeds the threshold[7]. Here, each extracellular electrode serves as an ideal point source, with a fixed distance from an infinitely long, uniform axon[50]. The level of membrane polarization and polarity will vary in response to the axonal stimulation. For example, in an anodic stimulation, strong hyperpolarization arises in the membrane segment near the electrode, while weak depolarization arises at the distal end owing to the reverse electric field. For the electrical stimulus of the retina, a complex neural network, redistribution of charge across the membrane of soma, axons, and dendrites would all contribute to the depolarization of the retinal neurons. At the subcellular level, an action potential initiated on one neuronal component may propagate to another neuronal component, thereby substantially affecting the temporal and spatial response dynamics of the neuron cells[7,51].

The cytocompatibility of our device was investigated through cell viability testing. Prostate cancer (PC-3) cell line was cultured on the F-URSP for 1 and 3 days for analysis, with a control group of PC-3 cell line cultured on petri-dish substrates as reference. As shown in Supplementary Fig. 22a, the fluorescence microscopy staining imaging of viable and dead cells shows that the cells on the device present a similar morphology and density to the reference cells. The PC-3 cells exhibit comparable viability (>96.6%) to the control group on day 3 (Supplementary Fig. 22b), indicating that the F-URSP does not exhibit cytotoxicity (for detailed cell information, see "Methods"). In addition, the temperature increase of human eye tissue exposed to US was evaluated via FEA simulation (trigger parameters: 10% duty cycle, 1000 cycles per pulse, 50 Vpp) (Supplementary Fig. 23). The eyeball was simplified to four main parts: cornea, lens, vitreous body, and retina, where the shape and size of each part were preset[52]. Acoustic and thermal specifications of human eye tissues were obtained from the published literature[53]. The results indicate that the US-induced temperature increase is <0.5 °C. Most of the increased heat occurs within the lens because of the high acoustic attenuation of the lens (detailed parameters are listed in Supplementary Table 4). The temperature increase in eye tissue was experimentally measured (~0.4 °C) by inserting the implant into an excised porcine eyeball (Sierra Medical Science, Inc., Whittier, CA, USA) exposed to US at the same time-averaged acoustic intensity. Consequently, the F-URSP is ultrasonically safe to operate with average US intensity stimulation.

## Discussion

This study presents the design and implementation of a retinal prosthesis device for evoking phosphenes or artificial visual percepts to the blind through a wireless US-induced retinal stimulating piezo-array, expanding the functional scope of conventional ultrasonic electronics. This system leverages ultrasonic features, strategic material integration, advanced microprocessing technology, and sophisticated control algorithm to achieve the functionality that allows the acoustic field-induced acquisition of patterned vision. A US 2D array was used to dynamically generate different acoustic focusing patterns to excite the receiving pixels for electrical stimulation of retinal neurons. The area, distance, and intensity of the acoustic focusing can be flexibly adjusted through controlling the US input parameters to ensure that the main lobe can activate neurons, but sidelobes are weaker than the threshold to avoid out-of-focus activation. As a consequence, the pixels under US excitation produce sufficient electrical outputs (e.g., voltage ~0–4.3 V, power density ~0–22.6 mW cm$^{-2}$) that are higher than the stimulus parameters (e.g., an electrode voltage ~0.8 V) used by an EM-based commercially available visual prostheses[43]. US-induced electrical stimulation elicits the same neuronal response as conventional electrical stimulation methods. For every 0.3 ms of stimulation, the neuronal spike lasts about 10 ms; for the evoked calcium signal, it can last up to 30 s. In addition, the 2D array can dynamically generate different patterns in a frame rate higher than 1 kHz. Conventional electrical stimulation studies found that neuron responses have their own limitations. For example, Argus II, the existing commercial retina prosthesis, only works at 10 Hz frame rate while its device supports a frame rate higher than 120 Hz[11]. Therefore, the stimulation frequency of our ultrasonic visual prosthesis can be adjusted according to the actual demand.

Compared with current EM technologies, the US-induced capabilities would distinctly benefit from three main advantages: (1) a higher FDA limit for power flux (720 mW cm$^{-2}$ versus 10 mW cm$^{-2}$) that enables US devices to be more biosafe[20,54]; (2) higher security in cybercommunications, enabling low probability for someone to hack into an US machine[55]; and (3) a much smaller wavelength (five orders of magnitude shorter at a similar frequency) that allows the beam to be focused to a small spot (e.g., ~0.1–1 mm @ 1–10 MHz) for distributed stimulators and better coupling to tiny implant units (e.g., millimeter or submillimeter units)[21,24]. The efficiency of EM power transfer highly depends on the dimension of receiver coils. Although some studies have shown that EM transfer efficiency can reach 50%, power transfer efficiency for devices with receiver size <1 mm$^2$ significantly reduced (<2%)[56]. As a comparison, the acoustic conversion efficiency can reach 20.4% for an ultrasonic implant with a 750-μm$^3$ piezoelectric cubic receiver[24]. Therefore, the US-induced device demonstrates a promising strategy for biomimetic visual prostheses with volume efficiency and structural simplicity compared to EM alternatives with low efficiency and security risks. The fundamental technological achievements in this study provide opportunities for the next generation of artificial vision prostheses.

Nevertheless, device performance and functionality could be further improved toward translational applications. First, the acoustic-electrical coupling efficiency of the piezo-layers could be enhanced via material composition and structural design[57,58]. Higher electrical outputs could be achieved under lower acoustic intensities, thereby ensuring stimulation studied well below the safety limit and minimizing the ultrasonic biological effects (e.g., thermal and mechanical effects)[59]. More rigorous steps are needed for the long-term biocompatibility of the lead-based piezoelectric system in the eye in vivo. Second, an ultrathin, flexible or even stretchable format could be achieved through MEMS technology to provide comfortable mechanical compliance that allows seamless coupling with the retina[60–62]. Piezo-units and electrode arrays could be manufactured small (size <0.1 mm$^2$) and dense enough to enable placing orders-of-magnitude more pixels (hundreds or even thousands) into a device, providing high resolution and covering a sufficiently large region of the visual field[3]. Third, a high-frequency 2D array with larger channel counts could also be exploited to focus and steer the US beam with higher resolution and more refined shapes. For eventual use, the 2D array would be integrated into wearable smart glasses, with a flexible 3D-position adjustable design. An automatic calibration procedure would also be developed to assess the US imaging of the device and adjust the relative position in real-time to ensure efficient emission and reception of the US signal. A solid gel like an Aquaflex gel pad (an aqueous, flexible US scaffold) would be customized in a configuration to closely match the 2D array (eyeglasses) and the receiving piezo-array (eyeball surface) to prevent sloshing around and falling out with patient motion, thus enhancing its practicality. The F-URSP would be surgically sutured onto the sclera of the eyeball to hold the implant securely onto the eyeball surface and convert the transmitted ultrasound into electricity for stimulation. In addition, lens extraction (as performed routinely in cataract surgery) at the time of device implantation would be more advantageous considering that the lens will both attenuate and refract (defocus) the acoustic beams. Fourth, the biphasic stimulation solution would be adopted by incorporating a custom application-specific integrated circuit (ASIC) consisting of power management, control, and stimulation modules into the ultrasonic device for translational research, rather than monophasic stimulation that cannot achieve charge balance[63]. Last but not least, integrating post-end functionalities such as a micro-camera for videography and an IC for optical picture-to-program processing, signal processing, data communications, and advanced stimulation algorithms in a closed-loop control system would also significantly enhance device practicality[64]. These directions seem promising to continue research to unleash capabilities enabled by retinal stimulating piezo-array.

## Methods

**Fabrication of the F-URSP**. The fabrication process can be summarized into three parts: (1) design and preparation of the flexible PCB; (2) fabrication of the piezo-composite microarray; (3) assembly and encapsulation. First, according to the device design, a PCB layout was created using PCB design software. The PCB design was then fabricated using the commercially available PCB manufacturing service. Second, a dicing-and-filling technology (Tcar 864-1, Thermocarbon, Casselberry, FL, USA) was used to manufacture the PMN-PT crystal/epoxy 1-3 composites with miniature piezo-pillars (length: 90 μm, width: 90 μm) and kerfs (30 μm). The kerfs were filled with an insulating polymer (EPO-TEK 301 epoxy resin). The manufactured piezo-composites were then thinned to 180 μm and sputtered with Cr/Au (50/100 nm) electrodes. A mixture containing silver powder and epoxy resin was cured on the top side as a matching layer (35 μm). Next, the entire acoustic stack was diced into small blocks with a size of 8.5 mm × 8.5 mm, in which the footprint of each piezo-element is 1 mm × 1 mm and the gap is 0.5 mm. Third, the prepared composite component was bonded on the PCB by using conductive silver paste. After curing, a 10-μm Parylene-C layer was uniformly deposited over the device (except for the stimulating electrodes) to provide insulation protection and biocompatible passivation. Poling of the piezo-components was conducted at 2 V μm$^{-1}$ (d.c.) for 20 min.

**Material and output performance characterization**. The piezoelectric coefficient $d_{33}$ was evaluated using a $d_{33}$ meter (YE2730A, APC International Ltd., Mackeyville, PA, USA). An inductance-capacitance-resistance digital bridge instrument (1715 LCR, QuadTech, Inc., USA) was employed to measure the capacitance. The impedance and phase were measured using an impedance analyzer (4294A, Agilent, USA). Based on the IEEE standard[65], the permittivity $\varepsilon$, electromechanical coupling factor $k_t$, acoustic velocity $c_p$, and acoustic impedance $Z_a$ were calculated (Supplementary Table 2).

The US-induced electrical outputs of the F-URSP were measured using a multifunctional US testing platform. US transducers as the external acoustic sources were mounted on a 5-axis motorized stage. F-URSP as a US receiver was placed in front of the transducer. Degassed US gel (EcoVue US Gel, HR Pharmaceuticals, Inc., USA) was applied as the coupling medium between

transducer and biomimetic visual prostheses for transmitting US waves. The US transducer was driven via a tone sine-wave burst, which was generated by a function generator (AFG3252C, Tektronix, USA) and then amplified by 40 dB with an amplifier (75A250A, AR RF/Microwave Instrumentation, USA). An oscilloscope (TDS 5052, Tektronix, USA) with an internal resistance of 1 MΩ was used to measure the output voltages generated by the device. The acoustic pressure of the ultrasonic transmitter was measured in the water tank by a hydrophone probe (HGL-1000, ONDA, Inc., Sunnyvale, CA, USA).

**Pattern reconstruction**. First, the device was laminated on hemispherical support and a focused single-element transducer that was designed and fabricated in our laboratory by using a focused PZT ceramic disc with a diameter of 30 mm, a center frequency of 3.3 MHz, and a focal length of 25 mm (photograph and detailed pulse-echo test of the focused transducer are shown in Supplementary Fig. 5) was used to generate an active area by moving in a programmed path. The output voltage magnitude of each pixel was measured constantly through the fast-testing pixel by pixel. The image was reconstructed in a time domain by integrating the collected data. Second, a 2D US array transducer with a 3.5-MHz center frequency was used to emit complex acoustic fields. The Verasonics system (Verasonics release 4.2, Verasonics, Inc., USA) was used to drive the 2D array, and a customized script was used to electronically adjust the steering angle and focal distance in real-time. In the above two experiments, a tone sine-wave burst with duty cycle of 10% at 50 Vpp was used to drive the transducers. The acoustic field transmitted from the 2D array was measured in the water tank by a hydrophone probe (HGL-1000, ONDA, Inc., Sunnyvale, CA, USA).

**Study of living retina stimulation**. The retina was dissected from the Ai95(RCL-GCaMP6f)-D transgenic male mice (at 12–14 weeks of age) (The Jackson Laboratory). The sensitivity and kinetics of GCaMP6f make the Ai95(RCL-GCaMP6f)-D mice a preferred choice for long-term cellular imaging of neuronal populations. Ai95(RCL-GCaMP6f)-D mice showed higher cortical expression levels and it is stable over time (longer than 5 months)[66]. The isolated retina was put on the surface of the sterilized stimulation electrodes and immersed in the tissue medium. A sterilized glass sheet was put on the device to prevent the retina from floating away from the electrodes. All procedures were conducted at room temperature. All procedures involving mice were approved by the Institute for Animal Care and Use Committees of the University of Southern California (protocol number: 20978, USC).

Calcium imaging was conducted using the Leica MICA microscope in widefield mode, HC PL Fluotar 1.6x/0.05NA objective, and 470 nm LED light source. Imaging was conducted with 25 Hz frame rate. The US conditions for the living retina stimulation were at a transmitter trigger voltage of 50 Vpp, with the duty cycle of 10% and 1000 cycles per pulse, the corresponding spatial peak temporal average intensity ($I_{SPTA}$) of 0.606 W cm$^{-2}$, and the corresponding mechanical index (MI) of 0.238. The thermal effects of US under this condition were further simulated and measured to demonstrate the safety of US stimulation.

**Study of cytocompatibility**. Prostate cancer cell (PC-3) lines were purchased from American Type Culture Collection (ATCC) for the study of cytocompatibility. PC-3 lines were cultured and maintained in the complete growth medium with the following ingredients (volume ratio): 89% Gibco Dulbecco's modified Eagle's medium (DMEM) (Life Technologies Co., Grand Island, NY, USA), 10% Fetal Bovine Serum (FBS) (Sigma-Aldrich, St. Louis, MO, USA), and 1% antibiotics Penicillin-Streptomycin (PS) (Sigma-Aldrich, St. Louis, MO, USA). PC-3 lines were divided and cultured into 30 mm petri-dishes with 2 mL complete medium for 3 days in a $CO_2$ incubator (HERAcell 150i, Thermo Scientific, Waltham, MA, USA) under a condition of 37 °C and a $CO_2$ level of 5%. The ethanol-sterilize F-URSP was prepared and placed on the bottom of the petri-dishes to perform the cell culture experiment for 3 days. Meanwhile, the control groups of the petri-dishes were also cultured for 3 days as reference. Each group has five samples. LIVE/DEAD Cell Imaging Kit (Catalog No. R37601, Invitrogen, Thermo Fisher Scientific, Waltham, MA, USA) was used for live/dead staining. Two reagents are included in the LIVE/DEAD Cell Imaging Kit. Calcein AM (in green, excitation: 488 nm and emission: 515 nm) stains the live cells, and BOBO-3 Iodide (in red, excitation: 570 nm and emission: 602 nm) stains the dead cells. After preparation, the cells were imaged under fluorescence microscopy (Mode No. SPE, Leica Microsystems, Inc., IL, USA).

**Finite element analysis (FEA)**. FEA simulations were carried out using a Field II program[67,68] (an acoustic simulation tool based on MATLAB, R2019b, MathWorks, Inc., USA) and COMSOL software (COMSOL Multiphysics 5.3a, COMSOL, Inc., USA) to simulate the acoustic field distributions of the US transmitters and the corresponding US-induced piezo-potential distributions in elements, respectively. In acoustic field simulations of a 2D array transducer, a backpropagation method, angular spectrum method[69], was used to determine the amplitude and phase distribution on the 2D array for either letter. A three-dimensional US field generated by the US array was simulated using Field II. For better 3D image visualization, 3D volume data from Field II was rendered by Amira software (Thermo Fisher Scientific, Inc, MA, USA) to show a stereoscopic view. In COMSOL simulations, the physical domains considered include electrostatics, pressure acoustics (frequency-domain), and solid mechanics, as well as the coupled interfaces of the piezoelectric effect and acoustic-structural boundary. Material parameters were set according to supplier data sheets and experimental measurements. The geometries of the components were strictly in accordance with the designs.

**Statistics and reproducibility**. No statistical method was used to predetermine the sample size. No data were excluded. The experiments were randomized and investigators were blinded to outcome assessment.

**Reporting summary**. Further information on research design is available in the Nature Research Reporting Summary linked to this article.

## Data availability

The data that support the findings of this study are available from the corresponding authors upon reasonable request. The Source data generated in this study are provided in the Source data file. Source data are provided with this paper.

## Code availability

Numerical simulations in this work are performed using the commercial finite element software COMSOL Multiphysics and MATLAB. All related codes are available from the corresponding authors upon reasonable request.

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

## Acknowledgements

This work was supported in part by the National Institutes of Health (NIH) under grant no. R01 EY030126. G.L. is supported by the Alfred E. Mann Innovation in Engineering Doctoral Fellowship. We thank Jason Alexander Junge and the USC Office of the Provost and USC Translational Imaging Center at the University of Southern California for their support in the fluorescent imaging, and Nan Sook Lee at the University of Southern California for her assistance with the cytocompatibility test. We also thank the Leica-USC collaboration group, specifically Sebastian Tille and Patric Pelzer from Leica and Scott Fraser and Francesco Cutrale from USC, to provide the Leica widefield microscope.

## Author contributions

L.J. and Q.Z. conceived and designed experiments. L.J. prepared the piezo-composites, processed the F-URSP and transmitter fabrication, performed numerical simulations by PiezoCAD software and Field II, and conducted experiments, data collection, and analysis. G.L. performed physical modeling and simulations by Field II with MATLAB and COMSOL Multiphysics. L.J., G.L., J.B., and C.G. conducted the fluorescent imaging. Y.Z. performed ultrasonic output measurement and cytocompatibility test. Y.S. contributed to the dicing-and-filling technology and 2D array fabrication. H.K. contributed to the PCB design. M.S.H. provides assistance in the analysis of retina stimulation results. L.J. wrote the manuscript. L.J., Y.C., and Q.Z. supervised the work. All authors discussed and commented on the manuscript.

## Competing interests

M.S.H. has gotten royalty payments from Second Sight Medical Products and is an equity holder in Golden Eye Bionics. The authors declare no competing interests.
