## [Peer Review File · Nature Communications]

Flexible ultrasound-induced retinal stimulating piezo-arrays for biomimetic visual prosthesesEditorial Note: Parts of this Peer Review File have been redacted as indicated to remove third-party material where no permission to publish could be obtained.

REVIEWER COMMENTS

Reviewer #1 (Remarks to the Author):

The article presents results on a retinal prosthesis using ultrasound power transfer to power and send signals to a flexible, 32-channel stimulation electrode. The implantable device consists of a flexible polyimide PCB with piezoelectric receivers in a 32-channel array attached to the front of the eye and electrically connected to stimulation electrodes that contact the retina. When activated by an external 2D ultrasound array, the piezoelectric receivers generate an AC signal that is converted to a DC signal and passed to the stimulation electrodes at thresholds high enough to activate optical nerves. Using the multi-channel design the stimulation can be spatially targeted based on which elements of the array are excited.

The authors present a 1-3 piezoelectric composite receiving element, composed of lead magnesium niobate lead titanate (PMN-PT), with improved piezoelectric performance over bulk piezoelectric material and provided insight and guidelines to guide future designs. The results presented include optimization simulation and testing characterization of the receiver element (voltage efficiency $\sim 11.8\%$, power density 7.2 mW/mm^2 under 220Ω load and 50 Vpp trigger voltage). The authors provide comparisons for required electrical stimulation charges for other chronic retinal implants and demonstrate ability to evoke stimulation response based on reported literature values.

Using the presented receiving element, an ultrasound powered, flexible retinal prosthesis with 32-channel stimulation electrodes activated and controlled by integration of a 32-element piezoelectric receiver array is demonstrated. Stimulation was achieved via selective excitation of receiving elements which result in spatially coordinated stimulation of retinal nerves. These results are confirmed *ex vivo* where excitation of excised mouse retinas is demonstrated using the device. Finally, basic cytocompatibility is demonstrated using live dead cell culture in comparison with a control.

The device presented represents a strong example for application of wireless ultrasound power transfer technology to the field of implantable medical devices. The work innovates in the field of retinal prosthesis by demonstrating an integrated wireless implant and removes the need for connecting wires present in some early devices. Further development of the technology could enable miniaturization of functional components by leveraging the shorter wavelengths and increased power flux allowed for with ultrasound power transfer methods when compared to existing examples of electromagnetic strategies. The work expands on developments in the literature including *ex vivo* demonstration with live retinal nerve responses.

The work adequately supports its claims. But the claim for biocompatibility could be revised more accurately to say that the device is cytocompatible. Biocompatibility is a spectrum of response, and neither acute or chronic biocompatibility is evaluated.

The methodology is sound and meets standards in the field.

Overall, the methods are detailed enough to reproduce the work given clarification of the following:

The model of the focused transducer used for ultrasound delivery in the Pattern Reconstruction method section is not listed.

In the methods for: Study of Living Retina Stimulation, the ultrasound conditions are not described.

In figure 2 e, units have lower-case pa instead of Pa. Please change to Pa to ensure we know you are indicating microvolts per Pascal.

Is it possible for the authors to report delivered acoustic intensity at the face of the 32-channel receiving array and at the face of the transducer alongside the V_{pp} reported values?

Why prostate cancer cell lines to determine cytocompatibility? These are not particularly relevant, and most likely overestimate the cytocompatibility.

Reviewer #2 (Remarks to the Author):

This report describes a novel approach to artificial vision in cases where retinal photoreceptors are non-functional. The approach utilizes an ultrasound array on the surface of the eye which, when excited, produces sufficient charge to excite intact retinal nerves to generate a pattern that would be perceived as a visual image.

A significant and novel aspect of the report is that the ultrasound array when exposed to a pattern of focused acoustic waves would generate a corresponding electrical pattern on the retina, and this is demonstrated by exposing the array to 3.5 MHz focused ultrasound generated by single-element and 2D arrays. A shortcoming of the paper is that it is not explained how an ultrasound array mounted on eyeglasses would be able to stimulate the piezo array on the ocular surface in the absence of a fluid acoustic propagation medium, nor how the array mounted on the eyeglasses would translate an optical view into an acoustic pattern to stimulate the piezo array on the surface of the eye.

Line 34: The statement "There are still few treatment options for millions of patients with diseased or damaged eyes leading to blindness" is overly general. It would be better to be more specific by stating that there are few treatment options for patients with retinal photoreceptor degeneration.

Line 57: "720 mW·cm⁻²" This figure refers to maximum allowable temporal average intensity in ultrasound diagnostic systems for peripheral vessels. The allowable limit for the eye is much lower. However, given that the proposed system for retinal stimulation is not diagnostic, the relevance of this number is questionable. It would be of greater relevance to know the temperature increase in tissue with the implant present when exposed to ultrasound at the time-averaged acoustic intensity. Note that the piezo array would be mounted on the cornea, which lacks blood-flow.

Line 98: "...if their optic nerve is intact." This presupposes that the retinal ganglion cells are also intact and functional.

The device design as described is confusing (at least to this reviewer). The eyeglasses seem to incorporate an ultrasound array and 'beamlines' are said to stimulate the piezo array on the surface of the eye which then excites the implanted retinal electrode array. But if the eyeglass 2D array is not in contact with the eye, how can acoustic 'beamlines' transmit through the air to excite the piezo array? Also, how is optical information encoded by the 2D eyeglass array without a system for optically focusing the image upon the array elements?

Fig 1b appears to show the piezo array seated on the cornea with flexible conducting path to the electrical array to be implanted on the retina. Must the eye be surgically opened so that the electrode array sits on the retina? Or would the array be inserted behind the choroid or RPE or sclera?

Fig 3 demonstrates conversion of ultrasound to voltages relevant for retinal stimulation. How do these voltages compare with those used in the Argus II?

The authors should briefly explain the rationale and properties of the Thy1-GCaMP6f transgenic mice for fluorescence imaging.

Fig 6 shows retina pre/post stimulation. There appear to be two small spots of fluorescence in the excited image. Do these correspond spatially to the expected excitation positions? How reproducible was this?

The chief strength of this report is demonstration of conversion of ultrasound energy into voltages that might be used to stimulate retina and details of its fabrication. The demonstration of retinal excitation-induced fluorescence is not convincing as this seems to be an n of 1 and we do not know if the two points showing fluorescence correspond spatially to the expected position or positions of excitation. Generation of a V or H pattern as in Fig 5 would have been more convincing.

While a chief point of the report is non-contact stimulation, it is not explained how focused ultrasound beams generated from the array on the eyeglasses would travel to the surface of the eye to stimulate the implant in the absence of a fluid propagation medium.

Reviewer #3 (Remarks to the Author):

See attached

Overall, this is an impressive technologically advanced work and presented with a lot of design rigor and modeling and simulation results. However, the experimental results are unimpressive and therefore does not take this paper to the level of acceptance by Nature Comm. With stronger experimental evidence, new and convincing data (eg Fig. 6 analogous to simulation Fig. 5) the paper holds potential for future acceptance.

Review summary

This paper presents an impressive ultrasonic transducer array, use of cutting edge microfabrication to build well-matched arrays to produce signals, focusing, and achieving powering to stimulate retina. The paper, therefore, address the need for novel approach to retinal stimulation that is not limited by electrical stimulation wiring, current density, and power delivery options. In this work, the transducer array shows excellent spatial resolution in information transfer in visual prostheses. Overall, the paper is well organized and exciting to read for its technological innovation. A new approach for retinal slice system and testing of the device is presented as a precursor for future in vivo solutions.

1. The major strength of the paper is the design, innovation: paper has extensive design information, fabrication is impressive, and modeling is highly supportive of the design and stimulation produced by the flexible US array stimulation method. The strength is the state-of-the-art fabrication, design and demonstration of the device through modeling and simulation data. Several comments are made below, and should be addressed and will clarify and enhance the paper.
2. (a) First, main weakness is the retinal cell/slice data is quite limited; only pixelate stimulation and 3 day relative equivalence are provided. This is a serious weakness; the pixelated stimulation in Fig. 6 c, d is not impressive at all. Indeed, there is no demonstrated focal stimulation of the retina, and none of the features of Fig. 5 are reproduced.

To be honest, the photo in Fig. 6b isn't helpful. Can you improve angle/clarity/labelling?

- Can you explain many bright pixels in c,d since the pixels of interest in 2 d,e
- Fig. 6 f-g are nice tutorial, idealization but not data
- But what matters are traces in 6i and they are not well explained.

2. (b) The second major limitation is that no in vivo biocompatibility results are provided (even though mentioned earlier on). 3 days is hardly a biocompatibility test (more a test of retinal slice health). This issue/biocompatibility is tangential and should be removed.

- Figure 6i: Why did you use a prostate cancer cell line? This cell type is completely different from cells found in the retina from both hardness and proliferability standpoints. It would be much more convincing to see this experiment done on RPE cells, or better yet, *in vivo*. Until then, please reduce the claim made on line 313 as to the "excellent biocompatibility."
- Also on line 313-314: there are no data on the cell culturing process not changing device performance. Please remove this claim or provide data.

Initial steps of biocompatibility were demonstrated through 3 days of cell culture work. For translational work, more rigorous steps will likely be needed for a lead-based piezoelectric system for the eye.

3. There are some design and technological issues not covered:
 - Other factors are speed of writing as shown in Fig. 5
 - How rapidly are the pixels activated? Is there a focus, spillover/outside of the focus as in Fig. 5?
 - What is the temporal persistence?

- No action potential or EPSP or any such excitability Df fluorescence traces are provided
 - How does the F-URSP function over time (diminishing/altered output)? This device is supposed to act as a visual restoration aid, so how does the system function over the period of minutes or days? In what conditions does the F-URSP work best?
 - What is the protocol for spatial alignment of the 'transducer-receiver' both in this experiment and for eventual use? Glasses, piezo-array, and eyes themselves are dynamic with respect to each other with the potential to affect the received ultrasound signal. Such practicalities should be discussed.
4. Further, the discussion does not include the power, focusing and other efficacy issues or comparison with electrical or optogenetic alternatives. Some descriptive advantages of US are mentioned in the introduction but not in discussion: the system size, total power and complexity are not mentioned.
- Discussion lacks references, eg for “Furthermore, an ultrathin and stretchable format could be achieved through MEMS technology to provide comfortable mechanical compliance that allows intimate coupling with the retina.”
 - Supplementary Table 1 mentions the favorable comparisons but not the critical ones of focusing, speed of scan and pattern (how rapid, frame-rate, etc) stimulation threshold, and secondary effects (such as electrical dc, vs biphasic).
 - Table 3 gives several electrical alternatives, but not US-based alternative in comparison.

Other topics for producing improvement to the paper.

The abstract exceeds the suggested 150-word limit.

p.3, l.70: The command signals can be appropriately coded and wirelessly transmitted through an external ultrasound transducer, such as a 2D array

- ➔ Clarify this sentence; signal is transmitted, meaning signal/power...or you mean stimulation generated?
- ➔ Is 2D transmitter, receiver array same resolution, 1-to-1 matched? Fig. 1 shows 1-256 -> going to 1-32. So this aspect ratio is not clear.

efficiency was obtained in each 1-3 piezo-element

- Pertains to above: 1-3 composite array: what about the receiving elements?
- Aspect ratio in Fig. 1 is not clear

p.3, l. 78, and its biocompatibility ... this is an awkward sentence; should be broken into two parts, on what/why is biocompatibility.

p.3 l. 94; converted into a DC signal by the integrated rectifiers => generally electrical stimulation is biphasic; so dc has to be explained and contrasted.

Line 104: I think you mean to refer to Fig 1b, not 2b.

p.4, l.124 designed and fabricated via a dicing-and-filling method to optimize the device performance (**Fig. 2a**) -> will you be presenting the methods or reference? Would be good to have dimensions on 2a.

None of Fig 1 inset, 3 a, 5a give dimensions, particularly z-axis to give an idea of the size/scale of the array. In one picture, the 2D array dimensions, to depth/z-axis range for focusing are not evident.

- One anticipated problem might be the curvature of the f-URSP; so not all pixels are at equal distance and therefore may not be focused enough or produce equivalent stimulation. But there are some serious limitations here.

Figure 1: The schematic does not include ultrasound gel between the wearable transducers and flexible PCB. This is slightly misleading as to the system's anticipated ultimate ease of use. Please add this to the figure and answer the question: what solutions will there be to the need for gel between the transducer and PCB?

Fig. 2 uses different terminology than Fig. 1

- The receiver array shows 32 pixels, but 4x4 (green -> are these 1x1 mm elements) and then even more MPN-PT rods; how many?
- 2h simulated potentials has no scale; I has 'normalized' scale for comparison, but value?
- Figure Two skips letters 'f' & 'g' (which may be intentional).

Fig. 3

- Can you justify, explain 3.3MHz frequency?
- The axial dimension to focusing is 2.5 cm; that seems high, but clarify that it is scalable, programmable
- Output voltage, its variability, scaling with different power, etc is clear. What's not clear is the stimulus threshold, and currents - it's the current that stimulates; so for the applied voltage, load determines the current: so, what is the expected load?

Fig. 4 very nicely illustrates the beam focusing. Importantly, the graphs show spillovers to nearby pixels. The SNR is reasonably apparent; but the dB scale given does not give an idea whether this will lead to retinal pixel stimulation. i.e. there should be some clarity on above/below (or graded) threshold for phosphene/retinal excitation.

Figure 5f-h: While the results are impressive, I do not agree on the clarity and reliability of the imaging results as claimed on lines 265-266. Please show multiple trials to demonstrate reliability in image reconstruction and provide quantitative measure of error between expected outcome and actual. For example, MSE per pixel between a simulated image reconstruction output and the results in 5f-h.

Supplementary Fig. 6 is very useful regarding focusing field; but it does not map into the field of stimulation of retinal cells. This can be obtained by combining the electrical and acoustic field calculations (supplementary Note 1): what power is needed for comparable focusing, field of stimulation

Dear Editor and Reviewers:

Thank you for your letter and for the reviewers' comments concerning our manuscript entitled "Flexible Ultrasound-Induced Retinal Stimulating Piezo-Arrays for Biomimetic Visual Prostheses" (Research Article, NCOMMS-21-42661). These comments and suggestions are all valuable and very helpful for revising and improving our manuscript. We have carefully studied the comments, performed supplementary experiments and simulations, given more discussion and explanation, added relevant references, and tried our best to correct and re-edit the manuscript and the supplementary information that we hope to meet with approval. The main modifications are highlighted in the revised manuscript, and the point-to-point responses to the reviewers' comments are listed as follows:

Reviewer #1 (Remarks to the Author):

The article presents results on a retinal prosthesis using ultrasound power transfer to power and send signals to a flexible, 32-channel stimulation electrode. The implantable device consists of a flexible polyimide PCB with piezoelectric receivers in a 32-channel array attached to the front of the eye and electrically connected to stimulation electrodes that contact the retina. When activated by an external 2D ultrasound array, the piezoelectric receivers generate an AC signal that is converted to a DC signal and passed to the stimulation electrodes at thresholds high enough to activate optical nerves. Using the multi-channel design the stimulation can be spatially targeted based on which elements of the array are excited.

The authors present a 1-3 piezoelectric composite receiving element, composed of lead magnesium niobate lead titanate (PMN-PT), with improved piezoelectric performance over bulk piezoelectric material and provided insight and guidelines to guide future designs. The results presented include optimization simulation and testing characterization of the receiver element (voltage efficiency ~11.8%, power density 7.2 mW cm^{-2} under 220Ω load and 50 Vpp trigger voltage). The authors provide comparisons for required electrical stimulation charges for other

chronic retinal implants and demonstrate ability to evoke stimulation response based on reported literature values.

Using the presented receiving element, an ultrasound powered, flexible retinal prosthesis with 32-channel stimulation electrodes activated and controlled by integration of a 32-element piezoelectric receiver array is demonstrated. Stimulation was achieved via selective excitation of receiving elements which result in spatially coordinated stimulation of retinal nerves. These results are confirmed ex vivo where excitation of excised mouse retinas is demonstrated using the device. Finally, basic cytocompatibility is demonstrated using live dead cell culture in comparison with a control.

The device presented represents a strong example for application of wireless ultrasound power transfer technology to the field of implantable medical devices. The work innovates in the field of retinal prosthesis by demonstrating an integrated wireless implant and removes the need for connecting wires present in some early devices. Further development of the technology could enable miniaturization of functional components by leveraging the shorter wavelengths and increased power flux allowed for with ultrasound power transfer methods when compared to existing examples of electromagnetic strategies. The work expands on developments in the literature including ex vivo demonstration with live retinal nerve responses.

The work adequately supports its claims. But the claim for biocompatibility could be revised more accurately to say that the device is cytocompatible. Biocompatibility is a spectrum of response, and neither acute or chronic biocompatibility is evaluated.

Authors' response: We thank the reviewer very much for carefully assessing our work and giving lots of constructive comments and suggestions needed to improve our manuscript. In the revised manuscript, the claim for biocompatibility has been revised to say cytocompatibility or that the device is cytocompatible.

- The methodology is sound and meets standards in the field.

Overall, the methods are detailed enough to reproduce the work given clarification of the following: The model of the focused transducer used for ultrasound delivery in the Pattern Reconstruction method section is not listed.

Authors' response: Thanks for the comment. The focused transducer used for ultrasound delivery is designed and prepared by our laboratory using a focused PZT piezoceramic disc with a diameter of 30 mm, a center frequency of 3.3 MHz, and a focus length of 25 mm. The optical photo and detailed pulse-echo performance of the focused transducer are shown in **Supplementary Figure 5**.

Following the reviewer's comment, details of focusing probes are listed in the Pattern Reconstruction method section in the revised manuscript.

- In the methods for: Study of Living Retina Stimulation, the ultrasound conditions are not described.

Authors' response: We appreciate the reviewer's comment. The ultrasound conditions for the living retina stimulation were at a transmitter trigger voltage of 50 Vpp (Sinusoidal signal: 10% duty cycle, 1000 cycles per pulse), with the spatial peak temporal average intensity (I_{SPTA}) of 0.606 W cm^{-2} , and the corresponding mechanical index of 0.238.

Following the reviewer's suggestion, the conditions have been listed in the Methods section in the revised manuscript.

- In figure 2 e, units have lower-case pa instead of Pa. Please change to Pa to ensure we know you are indicating microvolts per Pascal. Is it possible for the authors to report delivered acoustic intensity at the face of the 32-channel receiving array and at the face of the transducer alongside the Vpp reported values?

Authors' response: We are grateful that the reviewer has pointed out this question. We have modified it (pa → Pa) in the revised manuscript. Additionally, the spatial peak temporal average intensity ($I_{SPTA} = NPP^2/2\rho c \times \text{Duty cycle}$, where NPP is negative peak pressure, ρ is the density, and c is sound speed in the medium) at the face of receiving array alongside the Vpp was measured and plotted in **Supplementary Figure 11**. Here, the I_{SPTA} represents the focus area intensity at the face of receiving array. Since the acoustic beam is focused, the intensity at the face of receiving array (focus area) is, on average, 5.3 times the intensity at the face of transducer.

Following the reviewer’s suggestion, the acoustic intensity at the face of receiving array alongside the trigger voltage has been added in the revised manuscript (**Supplementary Figure 11**).

Supplementary Figure 11 | Spatial peak temporal average intensity (I_{SPTA}) at the face of receiving array alongside the trigger voltage. The I_{SPTA} represents the focus area intensity at the face of receiving array and is calculated by $I_{SPTA} = NPP^2/2\rho c \times \text{Duty cycle}$, where NPP is the negative peak pressure, ρ is the density ($\sim 1000 \text{ kg m}^{-3}$), and c is sound speed in the medium ($\sim 1540 \text{ m s}^{-1}$). The duty cycle used is 10%. Since the acoustic beam is focused, the intensity at the face of receiving array (focus area) is, on average, 5.3 times the intensity at the face of transducer.

- Why prostate cancer cell lines to determine cytocompatibility? These are not particularly relevant, and most likely overestimate the cytocompatibility.

Authors’ response: We appreciate the reviewer’s question. In the labs, many researchers usually use human cancer cell lines to study their characteristics in vitro, including biocompatibility and cytocompatibility. PC-3 cells are human prostate cancer cell lines, which are available. Hence we used Pc-3 cells for cytocompatibility. Moreover, in previously published works, such as:

[1] Sheng, H. W. et al. A thin, deformable, high-performance supercapacitor implant that can be biodegraded and bioabsorbed within an animal body. *Science Advances* **7**, eabe3097 (2021).

[2] Zhao, W. J. et al. Biocompatible and label-free separation of cancer cells from cell culture lines from white blood cells in ferrofluids. *Lab on a Chip* **17**, 2243-2255 (2017).

These authors used the human cancer cell lines (including PC-3 cells) to characterize the biocompatibility of the devices. These cancer cell lines come from humans and can be well cultured in petri dishes. Therefore, our PC-3 cell lines are appropriate for testing cytocompatibility.

Additionally, our piezo-array is targeted for retinal nerve stimulation, which needs to meet biosafety requirements, and therefore, experiments on cytotoxicity are important and necessary. For translational work, we will implement more rigorous steps to test the cytocompatibility and biocompatibility of the piezoelectric system for the eye in the future.

Reviewer #2 (Remarks to the Author):

This report describes a novel approach to artificial vision in cases where retinal photoreceptors are non-functional. The approach utilizes an ultrasound array on the surface of the eye which, when excited, produces sufficient charge to excite intact retinal nerves to generate a pattern that would be perceived as a visual image.

A significant and novel aspect of the report is that the ultrasound array when exposed to a pattern of focused acoustic waves would generate a corresponding electrical pattern on the retina, and this is demonstrated by exposing the array to 3.5 MHz focused ultrasound generated by single-element and 2D arrays. A shortcoming of the paper is that it is not explained how an ultrasound array mounted on eyeglasses would be able to stimulate the piezo array on the ocular surface in the absence of a fluid acoustic propagation medium, nor how the array mounted on the eyeglasses would translate an optical view into an acoustic pattern to stimulate the piezo array on the surface of the eye.

Authors' response: We thank the reviewer very much for carefully assessing our manuscript and giving lots of constructive comments/suggestions and discussions needed to improve our manuscript.

(1) In our work, we proposed an ultrasound-based approach to artificial vision in cases where a 2D array was designed and fabricated as an ultrasound transmitter that is mounted on the eyeglasses through a custom frame, as shown in schematic **Fig. 1a**. The flexible stimulating piezo-array as ultrasonic receiver will be surgically sutured onto the sclera of the eyeball to hold the implant securely onto the eyeball surface and convert the transmitted ultrasound into electricity for stimulation. For effective ultrasonic transmission and coupling, in our work, degassed ultrasound gel (EcoVue Ultrasound Gel, HR Pharmaceuticals, Inc.) was applied as the coupling medium between the 2D ultrasound array and the receiving piezo array for transmitting ultrasound. Ultrasound gel is a (typically water-based) medium that enables ultrasound waves to couple directly into the human body without signal attenuation and is confined between the 2D ultrasound array and the receiving piezo array by a custom frame in this work. For better

presentation of the device operation process, the ultrasonic gel as a coupling agent is shown in the revised schematic diagram (as shown in revised **Fig. 1a**).

(2) For future translational work, we will consider the integration of a micro-camera for videography and an integrated chip for optical picture-to-program processing in the eyeglasses in real-time. The processed signal will be used directly as the excitation signal of the 2D array for excitation to emit the desired ultrasound pattern for pattern stimulation. This process will be performed intelligently and in real-time in the future. In this work, the ultrasonic patterns (e.g., letters “H” and “V”) are pre-programmed in the MATLAB program by combining the 2D ultrasound array design and the relative ultrasonic algorithm. Then the 2D ultrasound array is accordingly triggered by a Verasonics system to emit the required ultrasonic pattern to achieve the stimulation effect (please refer to **Fig. 5a-e**).

Following reviewer comments, the above description and discussion have been added to the **Discussion** section of the revised manuscript to facilitate optimization of future studies.

Fig. 1 | Design structure and working principle of the F-URSP. a, Schematic diagram showing the flexible ultrasonic device, with key components and aspect ratios labelled.

Fig. 5 | Recognition function induced by programmable acoustic field of a 2D array transducer. **a**, Schematic diagram showing one working model of the F-URSP, in which a 2D-array ultrasound probe was used to excite programmable acoustic fields, with key components and aspect ratios labelled. **c**, Schematic diagram showing the ultrasound beam steering and focusing with a transducer array by modifying the transmitting amplitude and phase of each element. **d**, Simulated pressure magnitude distribution at the X-Z plane of a “H” pattern. **e**, Simulated 3D acoustic intensity distribution of a “H” pattern emitted by a 2D array transducer. Color bar indicates the normalized acoustic pressure.

- Line 34: The statement “There are still few treatment options for millions of patients with diseased or damaged eyes leading to blindness” is overly general. It would be better to be more specific by stating that there are few treatment options for patients with retinal photoreceptor degeneration.

Authors’ response: We appreciate the reviewer’s professional suggestion. The statement has been replaced by “There are few treatment options for patients with retinal photoreceptor degeneration” in the revised manuscript (lines 31-32).

- Line 57: “720 mW·cm⁻²” This figure refers to maximum allowable temporal average intensity in ultrasound diagnostic systems for peripheral vessels. The allowable limit for the eye is much lower. However, given that the proposed system for retinal stimulation is not diagnostic, the relevance of this number is questionable. It would be of greater relevance to know the temperature increase in tissue with the implant present when exposed to ultrasound at the time-averaged acoustic intensity. Note that the piezo array would be mounted on the cornea, which lacks blood-flow.

Authors’ response: We thank the reviewer for suggestions to address this issue. We conducted a FEM simulation to address this concern. Acoustic module and bioheat transfer module were used in this simulation. In the simulation setup, the eyeball was simplified into four main parts: cornea, lens, vitreous body and retina where the shape and size of each part were preset^[1]. Acoustic and thermal specifications of human eye tissues were obtained from the published literature^[2]. Detailed parameters are listed in the following **Supplementary Table 4**.

Simulation (stimulation parameters: 10% duty cycle, 1000 cycles per pulse, 50 Vpp trigger voltage of Sinusoidal signal) indicates that the ultrasound-induced temperature increase is less than 0.5 degrees. As shown in the **Supplementary Figure 23**, most of the increased heat occurs within the lens because of the high acoustic attenuation of the lens. In addition, the temperature increase in eye tissue was experimentally measured (~ 0.4 °C) by inserting the implant into an excised porcine eyeball (Sierra Medical Science, Inc., Whittier, CA, USA) exposed to ultrasound at the same time-averaged acoustic intensity.

Consequently, the piezo-array is ultrasonically safe to operate with average ultrasound intensity stimulation. Following reviewer’s comments and suggestions, we updated the results in the revised manuscript.

Supplementary Figure 23 | FEA simulation of the temperature increase of human eye tissue exposed to ultrasound. Simulation (parameters: 10% duty cycle, 1000 cycles per pulse, 50 Vpp) indicates that the ultrasound-induced temperature increase is less than 0.5 °C. Most of the increased heat occurs within the lens because of the high acoustic attenuation of the lens. The temperature increase in eye tissue was experimentally measured (~ 0.4 °C) by inserting the implant into an excised porcine eyeball (Sierra Medical Science, Inc., Whittier, CA, USA) exposed to ultrasound at the same time-averaged acoustic intensity.

Supplementary Table 4. Acoustic and thermal specifications of human eye tissues

Eye tissues	Density (kg m ⁻³)	Sound speed (m s ⁻¹)	Heat capacity at constant pressure (J kg ⁻¹ K ⁻¹)	Thermal conductivity (W m ⁻¹ K ⁻¹)	Attenuation (dB cm ⁻¹ MHz ⁻¹)
Water	1000	1500	4178	0.62	0
Cornea	1062	1586	4178	0.58	0.78
Vitreous	1005	1532	3999	0.6	0.01
Lens	1076	1647	3000	0.40	1.19
Retina	1034	1538	3680	0.57	1.15

References:

[1] Lozano, D. C. & Twa, M. D. Development of a Rat Schematic Eye From In Vivo Biometry and the Correction of Lateral Magnification in SD-OCT Imaging. *Investigative Ophthalmology & Visual Science* **54**, 6446-6455 (2013).

[2] Nabili, M., Geist, C. & Zderic, V. Thermal safety of ultrasound-enhanced ocular drug delivery: A modeling study. *Medical Physics* **42**, 5604-5615 (2015).

- Line 98: "...if their optic nerve is intact." This presupposes that the retinal ganglion cells are also intact and functional.

Authors' response: Thanks for the comment. Indeed, the retinal ganglion cells are required to be intact and functional, and the visual nerve is also intact if the ultrasound-induced visual prosthesis is to function as a visual reconstruction. Following the reviewers' comment, we revised the relevant statements (**.....of the blind if their retinal ganglion cells and optic nerve are intact and functional**) in the revised manuscript (line 99).

- The device design as described is confusing (at least to this reviewer). The eyeglasses seem to incorporate an ultrasound array and 'beamlines' are said to stimulate the piezo array on the surface of the eye which then excites the implanted retinal electrode array. But if the eyeglass 2D array is not in contact with the eye, how can acoustic 'beamlines' transmit through the air to excite the piezo array? Also, how is optical information encoded by the 2D eyeglass array without a system for optically focusing the image upon the array elements?

Authors' response: We thank the reviewer for the professional questions.

(1) As in the schematic diagram (**Fig. 1a**), the 2D ultrasound array as an ultrasound transmitter is incorporated into eyeglasses, and the receiving piezo array is on the surface of the eye. In our work, degassed ultrasound gel (acoustic impedance ~ 1.5 MRayls) (EcoVue Ultrasound Gel, HR Pharmaceuticals, Inc.) was applied as the coupling medium between the 2D

ultrasound array and the receiving piezo array for efficiently transmitting ultrasound to stimulate the elements. The ultrasound gel is confined between the 2D ultrasound array and the receiving piezo array by a custom frame (not shown in the previous schematic diagram). To better present the device operation process, the ultrasonic gel as a coupling agent is illustrated in the revised schematic diagram (revised **Fig. 1a**). In future translational applications, special flexible ultrasonic coupling patches would be customized with similar acoustic impedance and special configuration to closely fit the 2D ultrasound array (eyeglasses) with the receiving piezo-array (eyeball surface) for ultrasonic transmission and fixation of the device architecture.

(2) For the question of how optical information is encoded, please refer to the first response above. For example, as a solution for future translational work, we will consider the integration of a micro-camera for videography and an integrated chip for optical picture-to-program processing in the eyeglasses in real-time. The processed signal will be used directly as the excitation signal of the 2D array for excitation to emit the desired ultrasound pattern for pattern stimulation. In this work, the patterns (e.g., letter “H”) are programmed in the MATLAB program by combining the 2D ultrasound array structure and the ultrasonic algorithm to modify the transmitting amplitude and phase of each element, and then the 2D ultrasound array is triggered by the Verasonics system to emit the required ultrasonic pattern to achieve the stimulation effect.

Following reviewer comments, the above discussion has been added to the Discussion section of the revised manuscript in order to facilitate the optimization of future translational studies.

- Fig 1b appears to show the piezo array seated on the cornea with flexible conducting path to the electrical array to be implanted on the retina. Must the eye be surgically opened so that the electrode array sits on the retina? Or would the array be inserted behind the choroid or RPE or sclera?

Authors’ response: We thank the reviewer for the questions. Yes, minimally invasive surgery is required to implant the array on the retina. The surgery is well-established and safe. The surgery for our device can refer to the commercial Argus[®] II implant.^[1] Analogous to Argus II implant (refer to **Supplementary Figure 1**), our device integrates an ultrasonic receiving piezo-array and

a stimulating electrode array, and these components are supported on a flexible PCB with an intraocular connector to the epiretinal 32-electrode array (**Fig. 1a**). The flexible PCB connector is surgically sutured onto the sclera of the eyeball to hold the implant securely onto the eyeball surface. Standard vitrectomy surgery is performed, and then the electrode array is inserted into the vitreous cavity and positioned with the correct orientation on the macula. A titanium tack is then used to secure the heel of the implant to the retinal surface and hold the electrode array in position across the macula. As a reference, typical surgery times for commercial Argus® II implant are up to four hours, depending on the surgeon experience.

Reference:

[1] Ayton, L. N. et al. An update on retinal prostheses. *Clinical Neurophysiology* **131**, 1383-1398 (2020).

- Fig 3 demonstrates conversion of ultrasound to voltages relevant for retinal stimulation. How do these voltages compare with those used in the Argus II?

Authors’ response: Thanks for the question. For commercial products, Argus II (Argus® II Retinal Prosthesis (Implant), Second Sight Medical Products, Inc., Sylmar, CA 91342, USA) usually applies low electrode voltage (~ 0.8 V) and pulse amplitude of (~ 0-1000 μ A, including 4 current ranges: 0-125 μ A, 0-250 μ A, 0-500 μ A and 0-1000 μ A). As a comparison, the voltage and current outputs of our ultrasound-induced device can reach up to 4 V and 2000 μ A, respectively, thus ensuring that the device can stimulate under different conditions. The cited references are as follows (**Fig. R1**):

Editorial Note: © 2022 IEEE. Reprinted, with permission, from D. D. Zhou, J. D. Dorn and R. J. Greenberg, "The Argus® II retinal prosthesis system: An overview," *2013 IEEE International Conference on Multimedia and Expo Workshops (ICMEW)*, 2013, pp. 1-6, doi: 10.1109/ICMEW.2013.6618428.

Fig. R1. The voltage excursion waveforms recorded during long-term stimulation of a Argus II product (Zhou, D. D., Dorn, J. D. & Greenberg, R. J. The Argus® II retinal prosthesis system: An overview. In Proc. 2013 IEEE International Conference on Multimedia and Expo Workshops (ICMEW) (IEEE, 2013).

And the product manual is attached as follows:

[REDACTED]

Editorial Note: The redacted material here can be seen at Chapter 7, page 177 in the Argus® II Retinal Prosthesis System Surgeon Manual (Company & product: Second Sight Medical Products, Inc. 12744 San Fernando Rd., Building 3 Sylmar, CA 91342, USA & Argus® II Retinal Prosthesis (Implant)).

(Argus® II Retinal Prosthesis (Implant), Second Sight Medical Products, Inc, Sylmar, CA 91342, USA)

- The authors should briefly explain the rationale and properties of the Thy1-GCaMP6f transgenic mice for fluorescence imaging.

Authors' response: We thank the reviewer for the suggestion. The sensitivity and kinetics of GCaMP6f make the Thy1-GCaMP6 mice a preferred choice for long-term cellular imaging of neuronal populations. Thy1-GCaMP6 mice showed higher cortical expression levels than two GCaMP3 transgenic lines. Also, Thy1-GCaMP6 mice expression is stable over time (longer than 5 months)^[1]. Following reviewer comments, the explanation of Thy1-GCaMP6f transgenic mice was added to the Methods section.

Reference:

[1] Dana, H. et al. Thy1-GCaMP6 transgenic mice for neuronal population imaging in vivo. *PLoS One* **9**, e108697 (2014).

- Fig 6 shows retina pre/post stimulation. There appear to be two small spots of fluorescence in the excited image. Do these correspond spatially to the expected excitation positions? How reproducible was this?

Authors' response: Thanks for the questions. The small spots of fluorescence in the excited image are a local area. To demonstrate the ability of pattern stimulation, we conducted new calcium imaging experiments using isolated retina tissue and updated the results, as shown in **Figure 6b-d**. Using this piezo-array to stimulate the retina *ex vivo* under ultrasound excitation, the observed fluorescent area corresponds to the expected excitation location (pattern “V”), and the excitation phenomenon can be reproduced experimentally.

- The chief strength of this report is demonstration of conversion of ultrasound energy into voltages that might be used to stimulate retina and details of its fabrication. The demonstration of retinal excitation-induced fluorescence is not convincing as this seems to be an n of 1 and we do not know if the two points showing fluorescence correspond spatially to the expected position or positions of excitation. Generation of a V or H pattern as in Fig 5 would have been more convincing.

Authors' response: We appreciate the reviewer's professional comment and suggestion. The strength of our work is the demonstration of the original design and preparation of the ultrasound wirelessly induced piezoelectric array to achieve patterned stimulation in the retina. Therefore, to

demonstrate the ability of pattern stimulation more convincing, we performed new calcium imaging experiments *ex vivo* with isolated retina tissue. In this stimulation experiment, a two-dimensional V-pattern was emitted by using a two-dimensional ultrasound array transmitter, which was electrically triggered by a Verasonics system combined with the ultrasonic algorithm. Then, the correspondingly observed fluorescence-enhanced region corresponds to the expected excitation position (pattern “V”). Fluorescent intensity values were acquired from representative somas, which demonstrated an increase ($\Delta F/F_0 \sim 5\%$) in mean fluorescence intensity. It is a signal for the actual neural population activity. The results are updated in **Figure 6b-d**, which is an indication that ultrasonic pattern can be converted into an electrical pattern used to stimulate the retina for possible visual reconstruction.

Fig. 6 | Living retina stimulation and cytocompatibility of F-URSP. b, Simulated 3D acoustic intensity distribution of a “V” pattern emitted by a 2D array transducer. Color bar indicates the normalized acoustic pressure. **c,** Calcium imaging of the living retina at stimulated levels. **d,** Time course of calcium transients during stimulation.

- While a chief point of the report is non-contact stimulation, it is not explained how focused ultrasound beams generated from the array on the eyeglasses would travel to the surface of the eye to stimulate the implant in the absence of a fluid propagation medium.

Authors’ response: Thanks for the question. In the stimulation process, degassed ultrasound gel (acoustic impedance ~ 1.5 MRays) (EcoVue Ultrasound Gel, HR Pharmaceuticals, Inc.) was applied as the coupling medium between the ultrasound transducer and the surface of the eye through a custom frame for transmitting ultrasound to excite the implant. Ultrasound gel is a

(typically water-based) medium that enables ultrasound waves to couple directly into the human body, i.e. letting the waves transmit directly to the tissues that need to be treated or imaged. In future translational applications, special flexible ultrasonic coupling patches would be customized with similar acoustic impedance and special configuration to closely fit the transducer with the eyeball for ultrasonic transmission and fixation of the device architecture.

Following reviewer comments, we have given an explanation in the revised manuscript and more discussion on the devices toward translational applications was added in the Discussion section.

Reviewer #3 (Remarks to the Author):

Overall, this is an impressive technologically advanced work and presented with a lot of design rigor and modeling and simulation results. However, the experimental results are unimpressive and therefore does not take this paper to the level of acceptance by Nature Comm. With stronger experimental evidence, new and convincing data (eg Fig. 6 analogous to simulation Fig. 5) the paper holds potential for future acceptance.

Review summary

This paper presents an impressive ultrasonic transducer array, use of cutting edge microfabrication to build well-matched arrays to produce signals, focusing, and achieving powering to stimulate retina. The paper, therefore, address the need for novel approach to retinal stimulation that is not limited by electrical stimulation wiring, current density, and power delivery options. In this work, the transducer array shows excellent spatial resolution in information transfer in visual prostheses. Overall, the paper is well organized and exciting to read for its technological innovation. A new approach for retinal slice system and testing of the device is presented as a precursor for future in vivo solutions.

1. The major strength of the paper is the design, innovation: paper has extensive design information, fabrication is impressive, and modeling is highly supportive of the design and stimulation produced by the flexible US array stimulation method. The strength is the state-of-the-art fabrication, design and demonstration of the device through modeling and simulation data. Several comments are made below, and should be addressed and will clarify and enhance the paper.

Authors' response: We are very grateful to the reviewer for his/her careful assessment of our work and for his/her lots of constructive comments/suggestions and discussions needed to improve our manuscript.

2. (a) First, main weakness is the retinal cell/slice data is quite limited; only pixelate stimulation and 3 day relative equivalence are provided. This is a serious weakness; the pixelated stimulation in Fig. 6 c, d is not impressive at all. Indeed, there is no demonstrated focal stimulation of the retina, and none of the features of Fig. 5 are reproduced.

To be honest, the photo in Fig. 6b isn't helpful. Can you improve angle/clarity/labelling?

- Can you explain many bright pixels in c,d since the pixels of interest in 2 d,e
- Fig. 6 f-g are nice tutorial, idealization but not data
- But what matters are traces in 6i and they are not well explained.

Authors' response: Thanks to the comments.

(1) The strength of our work is the original design and preparation of F-URSP with wireless ultrasound transfer to achieve patterned stimulation in the retina. Therefore, to demonstrate the ability of pattern stimulation more convincing, we re-performed calcium imaging experiments *ex vivo* with living retina tissues dissected from Thy1-GCaMP6f transgenic mice. In the new stimulation experiment, a two-dimensional “V”-pattern was emitted by using the two-dimensional ultrasound array transmitter, which was electrically triggered by a Verasonics system combined with the ultrasonic algorithm. The correspondingly observed fluorescence-enhanced region on retina corresponds to the expected excitation position (pattern “V”). The results are updated in **Fig. 6b-d** in the revised manuscript, which is an indication that an ultrasonic pattern can be converted into a electrical pattern used to stimulate the retina for possible visual reconstruction.

(2) As suggested by the reviewer, we have replaced **Fig. 6b** with an improved angle, as shown in updated **Fig. 6a (right)** in the revised manuscript.

(3) The bright pixels represent fluorescence imaging Ca^{2+} transients of neuronal populations on the living retina. They are a signal for the actual neural population activity. Comparing before and after stimulation, the mean fluorescence intensity values of neuron cells increased under ultrasound-induced electrical stimulation, indicating enhanced activity of the actual neural population. Since the pixels are local and do not represent the features of **Fig. 5**, we re-performed calcium imaging experiments *ex vivo* with living retina tissues to achieve patterned stimulation.

As shown in **Fig. 6b-d**, the correspondingly observed -enhanced regions on retina correspond to the expected excitation positions (pattern “V”).

(4) Thanks to the reviewer. **Figs. 6f-g** illustrates the fundamentals of electrical stimulation to the retina. Combined with them, it is possible to help the reader understand the process of electrical stimulation. But to simplify **Fig. 6**, we moved it to the **Supplementary Material (Supplementary Figure 21)**.

(5) **Fig. 6i** (revised **Fig. 6e**) shows the fluorescent images of the PC-3 cell lines seeded on the petri-dish substrate and the F-URSP on day 3. The viable cells were stained in green, and the dead cells were stained in red, which shows that the cells on the device present a similar morphology and density to the reference cells. According to fluorescent images, we further analyzed cell viability, as shown in revised **Fig. 6f**. The results show that cells exhibit comparable viability (> 96.6%) to the control group, indicating that the F-URSP does not exhibit cytotoxicity. Following the reviewer's comment, we have re-edited the paragraph and revised **Fig. 6** in the revised manuscript.

Fig. 6 | Living retina stimulation and cytocompatibility of F-URSP. **a**, Schematic diagram and photograph showing the *ex vivo* retina tissue and experimental setup for the retina stimulation using the F-URSP. The retina tissue was removed from the eye of an anesthetized mouse and stimulated using F-URSP. The inset shows the living retina placed under the stimulating electrodes of the F-URSP. Scale bar: 1 mm. **b**, Simulated 3D acoustic intensity distribution of a “V” pattern emitted by a 2D array transducer. Color bar indicates the normalized acoustic pressure. **c**, Calcium imaging of the living retina at stimulated levels. **d**, Time course of calcium transients during stimulation. **e**, Cytocompatibility test: live and dead cell staining of the PC-3 seeded on the petri-dish substrate and the F-URSP on day 3. The viable cells were stained in green, and the dead cells were stained in red. Scale bar: 100 μ m. **f**, Cell viability of PC-3 cell lines seeded on the petri dish substrate and F-URSP, after 1 and 3 days of culture.

2. (b) The second major limitation is that no *in vivo* biocompatibility results are provided (even though mentioned earlier on). 3 days is hardly a biocompatibility test (more a test of retinal slice health). This issue/biocompatibility is tangential and should be removed.

Authors' response: We appreciate the reviewer's professional comment and suggestion. *In vivo* biocompatibility is indeed very important for the application of implantable devices. In view of this, our device has some considerations. First, a parylene-C film is uniformly deposited on the entire surface of the piezoelectric array (except for the stimulating electrodes) by chemical vapor deposition to provide insulation protection and biocompatible passivation. Parylene C is currently being used in many well-documented bio-medical implantable devices, and it has been proven to be a terrific biocompatible material. It is USP Class VI implantable plastic material and conforms to material ISO-10993 Biological Evaluations for Medical Applications. Parylene C is also probably the longest proven protective biocompatible material^[1].

Second, the structural design of the piezoelectric array device is based on the size of the human eye. To demonstrate the ability of stimulation more convincing, we performed calcium imaging experiments with an *ex vivo* retina dissected from Thy1-GCaMP6f transgenic mice. Such a device size is difficult to complete *in vivo* experiments in a mice. Please understand the difficult conditions of our experiments if larger-sized animals are to be used for this *in vivo* experiment. Nevertheless, new calcium imaging experiments *ex vivo* with isolated retina tissue were performed. The correspondingly observed fluorescent region corresponds to the expected excitation position (e.g., pattern "V"). The results are updated in **Fig. 6b-d**, which is an indication that an ultrasonic pattern can be converted into an electrical pattern used to stimulate the retina for possible visual reconstruction. Although we attempted to demonstrate biocompatibility by 3 days of cell culture, this was indeed limited. Additionally, according to the above reviewers' comments that biocompatibility could be revised more accurately to say that the device is cytocompatible, so we have re-edited it with using cytocompatibility to the relevant description in the revised manuscript.

We appreciate your comments and suggestions, which we have updated in the revised manuscript.

References:

[1] Stark, N. Literature Review: Biological Safety of Parylene C. *Medical Plastics and Biomaterials* **3**, 30-35 (1996).

- Figure 6i: Why did you use a prostate cancer cell line? This cell type is completely different from cells found in the retina from both hardness and proliferability standpoints. It would be much more convincing to see this experiment done on RPE cells, or better yet, *in vivo*. Until then, please reduce the claim made on line 313 as to the “excellent biocompatibility.”

- Also on line 313-314: there are no data on the cell culturing process not changing device performance. Please remove this claim or provide data.

Authors’ response: We thank the reviewer for the question and suggestions.

(1) PC-3 cells are human prostate cancer cell lines that come from humans. Currently, many researchers usually use human cancer cell lines to study their characteristics *in vitro*, including cytocompatibility/biocompatibility, such as in the published literatures:

[1] Sheng, H. W. et al. A thin, deformable, high-performance supercapacitor implant that can be biodegraded and bioabsorbed within an animal body. *Science Advances* **7**, eabe3097 (2021).

[2] Zhao, W. J. et al. Biocompatible and label-free separation of cancer cells from cell culture lines from white blood cells in ferrofluids. *Lab on a Chip* **17**, 2243-2255 (2017).

In these papers, the authors utilized the human cancer cell lines (including PC-3 cells) to study the cytocompatibility/biocompatibility of the devices. In addition, cancer cell lines can be cultured and grow well in Petri dishes. Thus, we use PC-3 cell lines to test the cytocompatibility of our device.

(2) For the line 313-314, we thank the suggestion from reviewer and have removed this claim (not changing device performance).

- Initial steps of biocompatibility were demonstrated through 3 days of cell culture work. For translational work, more rigorous steps will likely be needed for a lead-based piezoelectric system for the eye.

Authors' response: We thank the reviewer for the comment. In this work, our innovation focuses on the original design and preparation of a prototype device for a visual prosthesis that uses an ultrasound-induced piezo-array to stimulate the retina. Initial steps were demonstrated through 3 days of cell culture work to examine the device's cytocompatibility. For lead-based piezoelectric systems, previous researchers have reported utilizing lead zirconate titanate (PZT) materials as receiving elements to manufacture implantable devices for ultrasonic energy transfer. For example, Michel M. Maharbiz *et al.* reported a PZT-based implantable ultrasonic system to monitor deep-tissue oxygenation.^[1] For translational work, we will implement more rigorous steps to test the cytocompatibility and biocompatibility of the lead-based piezoelectric system for the eye in future, which was added in the Discussion section in the revised manuscript.

References:

[1] Sonmezoglu, S., Fineman, J. R., Maltepe, E. & Maharbiz, M. M. Monitoring deep-tissue oxygenation with a millimeter-scale ultrasonic implant. *Nat. Biotechnol.* **39**, 855-864 (2021).

3. There are some design and technological issues not covered:

- Other factors are speed of writing as shown in Fig. 5.
- How rapidly are the pixels activated? Is there a focus, spillover/outside of the focus as in Fig. 5?
- What is the temporal persistence?
- No action potential or EPSP or any such excitability Df fluorescence traces are provided.
- How does the F-URSP function over time (diminishing/altered output)? This device is supposed to act as a visual restoration aid, so how does the system function over the period of minutes or days? In what conditions does the F-URSP work best?

- What is the protocol for spatial alignment of the ‘transducer-receiver’ both in this experiment and for eventual use? Glasses, piezo-array, and eyes themselves are dynamic with respect to each other with the potential to affect the received ultrasound signal. Such practicalities should be discussed.

Authors’ response: We appreciate the reviewer’s comments and questions.

(1) Speed of writing is determined by two aspects: 1. Engineering part: ultrasound and electrical stimulation’s writing speed; 2. Biological part: retina neurons’ response rate. The engineering part has a very high speed of writing. Ultrasound 2D array can dynamically generate different patterns (as shown in **Fig. 5**) in a frame rate higher than 1 kHz. Then the ultrasound-induced electrical stimulation will stimulate the retina at the same frame rate. However, for the biological part, conventional electrical stimulation study found that neuron responses have their own limitation. For example, Argus II, the existing commercial retina prosthesis, only works at 10 Hz frame rate while its device actually supports a frame rate higher than 120 Hz. Many neuroscientists and researchers are trying to increase the potential frame rate by looking for more efficient electrical stimulation parameters and waveforms, but this is a neuroscience question, which is not the research focus of our study. Our device can always adopt the optimized electric parameters for stimulation since the system can produce a broad set of electrical stimulation parameters by adjusting the ultrasound input.

(2) In this study, we used ultrasound waves with 1000 cycles at 3.3-3.5 MHz for stimulation, which is ~ 0.3 ms, to activate the neurons. Indeed, there would be spillover/out of focus activation caused by the electrical stimulation. Two reasons can cause out-of-focus neuron activation: 1. sidelobes in the ultrasound beams. Usually, the ultrasound sidelobes are at least 10 dB weaker than the main lobe, so we can control the amplitude of ultrasound input, ensuring the main lobe can activate neurons but sidelobes are weaker than the threshold to avoid out-of-focus activation. 2. Electrical stimulation can sometimes cause widespread neuron responses (response area larger than the electrode size) when the ganglion cells’ axons are unintendedly activated. Recent research has found that longer stimulation pulses with a lower amplitude can confine the response area.^[2] Our device can always adopt the optimized electric parameters for retina stimulation since we can control the electrical stimulation parameters by adjusting the ultrasound input.

(3) Ultrasound-induced electrical stimulation causes the same neuron responses as conventional electrical stimulation methods. For each 0.3 ms stimulation, neuron spikes last for around 10 ms; For the evoked calcium signals, it can last for 30 seconds^[3].

(4) In this study, we use calcium imaging, a widely-used technology, to monitor the neuron activity. Genetically encoded calcium indicator (GCaMP6) responds to the binding of Ca^{2+} ions and shows fluorescence signals. Therefore, it can be used to monitor the neuron electrical activities. The trace of fluorescence change ($\Delta F/F$) is also shown in **Fig. 6d**.

(5) Our experiment lasted for several months in total, testing with various input power. No functional variation or diminished output was observed. The core composition of the F-URSP is the PMN-PT piezoelectric material, which has been used in the fabrication of commercial medical ultrasound devices for many years and be proved to be stable by the industry^[4]. Additionally, the Parylene-C film is uniformly deposited on the surface of the piezoelectric array to provide a protective layer. Parylene C is currently being used in many well-documented bio-medical implantable devices. Although we cannot test the device's stability in a natural human body for years, there is no reason to expect any diminishing performance.

The performance of F-URSP's (the core part is PMN-PT piezoelectric material) will be degraded or even damaged if the working temperature is higher than 80 °C (~ half curie point) because the PMN-PT has a curie point of 166 °C. However, this is not a possible temperature in practical implantation conditions. Additionally, no other concerns have been reported to affect the performance of PMN-PT in implantation. Therefore, in the case of implantation, the piezoelectric device will operate normally in the absence of destructive conditions.

(6) In this experiment, ultrasound imaging method was used to evaluate the spatial alignment of the 'transducer-receiver'. The transducer in our study is not only used for the emission of patterned acoustic fields but also for ultrasound imaging to evaluate the relative position of the receivers, as shown in **Supplementary Figure 19**. Depending on the results of ultrasound imaging, the position of the transmitter that is fixed on a 3D adjustable mount can be flexibly adjusted to align it with the receiver before the sound field is emitted. For future eventual use, the 2D transmitter will be integrated into wearable smart glasses with a flexible 3D-position adjustment design. An automatic calibration procedure would also be developed to

assess the ultrasound imaging of the receiver and adjust the position in real-time to ensure efficient emission and reception of the ultrasound signal.

Supplementary Figure 19 | The alignment of the “transducer-receiver” calibrated by ultrasound imaging. Additionally, the transducer is not only used for the emission of programmable acoustic fields, but also for ultrasound imaging to evaluate the alignment of the transmitter and receiver. Depending on the results of ultrasound imaging, the relative position of the transmitter that is fixed on the 3D adjustable mount can be flexibly adjusted to align it with the receiver prior to the emission of acoustic fields.

Following the reviewer’s comment and suggestion, these discussions have been supplemented in the revised manuscript to improve the paper.

References:

[1] De Bock, F., Dirckx, J. & Wyndaele, J. J. Evaluating the Use of Different Waveforms for Intravesical Electrical Stimulation: A Study in the Rat. *Neurourology and Urodynamics* **30**, 169-173 (2011).

[2] Weitz, A. C. et al. Improving the spatial resolution of epiretinal implants by increasing stimulus pulse duration. *Science Translational Medicine* **7**, 318ra203 (2015).

[3] Chow, A. Y. & Chow, V. Y. Subretinal electrical stimulation of the rabbit retina. *Neuroscience Letters* **225**, 13-16 (1997).

[4] Zhou, Q., Lam, K. H., Zheng, H., Qiu, W. & Shung, K. K. Piezoelectric single crystal ultrasonic transducers for biomedical applications. *Prog. Mater Sci.* **66**, 87-111 (2014).

4. Further, the discussion does not include the power, focusing and other efficacy issues or comparison with electrical or optogenetic alternatives. Some descriptive advantages of US are mentioned in the introduction but not in discussion: the system size, total power and complexity are not mentioned.

- Discussion lacks references, eg for “Furthermore, an ultrathin and stretchable format could be achieved through MEMS technology to provide comfortable mechanical compliance that allows intimate coupling with the retina.”

- Supplementary Table 1 mentions the favorable comparisons but not the critical ones of focusing, speed of scan and pattern (how rapid, frame-rate, etc) stimulation threshold, and secondary effects (such as electrical dc, vs biphasic).

- Table 3 gives several electrical alternatives, but not US-based alternative in comparison.

Authors’ response: We thank the reviewer for the comments and suggestions.

(1) Following the reviewer’s suggestion, the **Discussion** section has been rewritten in the revised manuscript, including more discussion on power, focusing, system size, efficacy issues, and the comparison with other electrical alternatives.

(2) In the **Discussion** section, we have added relevant references to the corresponding descriptions and discussions.

(3) **Supplementary Table 1** has been re-summarized by adding focusing, speed of scan, stimulation threshold, etc.

(4) Ultrasound-induced electrical stimulation has been reported for peripheral nerves, sciatic nerves, etc. However, the US-based electrical application for retina is our first proposal. Therefore, we re-edited **Supplementary Table 3** by giving several US-based electrical alternatives for peripheral nerves, and sciatic nerves to better understand and compare the ultrasound-based technique.

Other topics for producing improvement to the paper.

- The abstract exceeds the suggested 150-word limit.

Authors' response: We thank the comment from the reviewer. We have revised the abstract to meet the requirement of the 150-word limit.

- p.3, l.70: The command signals can be appropriately coded and wirelessly transmitted through an external ultrasound transducer, such as a 2D array

>> Clarify this sentence; signal is transmitted, meaning signal/power...or you mean stimulation generated?

>> Is 2D transmitter, receiver array same resolution, 1-to-1 matched? Fig. 1 shows 1-256 -> going to 1-32. So this aspect ratio is not clear.

Authors' response: Thanks for the questions.

(1) In this work, our piezo-device features the ability to pattern stimulation that can be achieved by ultrasonic fields. In this sentence that “the command signals can be.....”, we would like to express that the patterned stimulation can be achieved by controlled emission of the ultrasonic fields (distribution of acoustic power in space). The emission of the ultrasound fields was obtained by a 2D ultrasonic transmitter array that was controlled by the electrical signals transmitted by a Verasonics system. Therefore, it is possible to encode the electrical signals to control and trigger the transmitter to obtain different ultrasound fields (distribution of acoustic power in space) and thus achieve complex stimulation. Following the reviewer’s comment and suggestion, we have re-edited and updated the paragraph in the revised manuscript for the readers’ ease of understanding (page 3, lines 68-71).

(2) The 2D transmitter and the receiver array have different resolution and do not match one-to-one. The receiver array has 32 elements and each element size is $1 \times 1 \text{ mm}^2$. The whole size with the gaps is $8.5 \times 8.5 \text{ mm}^2$. The 2D transmitter is with 256 elements, and each element area is $0.66 \times 0.66 \text{ mm}^2$ (approximately equal to one and a half wavelength of the 3.5-MHz

ultrasound). The whole size of the 2D transmitter is $\sim 10.6 \text{ mm} \times 10.6 \text{ mm} \times 4.5 \text{ mm}$ (please refer to **Fig. 5a,b** for detailed structure and photograph). For pattern reconstruction, the resolution of the ultrasonic field is crucial, so here a transmitter with multiple and small array elements was designed. The resolution of the emission is higher than that of the receiver, so that the ultrasound field can reach the receiver surface with full and high accuracy. Following the reviewer's comment and suggestion, the detailed ratio parameters are given in the revised manuscript.

- efficiency was obtained in each 1-3 piezo-element
 - Pertains to above: 1-3 composite array: what about the receiving elements?
 - Aspect ratio in Fig. 1 is not clear

Authors' response: Thanks to the reviewer for your questions. In the piezo-array as a receiver (32 elements), each 1-3 piezo-element can be ultrasonically and individually activated. The whole receiving piezo-array with 32 elements (each composed of 1-3 PMN-PT/epoxy composite) and each element has a footprint of $1 \text{ mm} \times 1 \text{ mm}$ and a spacing of $500 \mu\text{m}$, as shown in **Fig. 2b**. The whole volume of the receiving piezo-array is about $8.5 \times 8.5 \times 0.18 \text{ mm}^3$ (length \times width \times thickness). **Supplementary Figure 2** shows the whole device design with detailed parameters. In addition, following the reviewer's comment, we have updated **Fig. 1** and given a detailed aspect ratio in the revised manuscript.

- p.3, l. 78, and its biocompatibility ... this is an awkward sentence; should be broken into two parts, on what/why is biocompatibility.

Authors' response: We thank the reviewer for the comment and suggestion. We have re-edited this sentence in the revised sentence, as follows:

“As a proof of principle, we demonstrated the continuous acoustic field-induced pattern reconstruction and ex vivo electrical stimulation response of the retinal tissue by using the F-URSP. To examine the device's cytocompatibility, we further cultured PC-3 cell lines on the substrates of the F-URSP. The PC-3 cell lines exhibit the viability ($> 96.6\%$) that is comparable

to the petri-dish control groups, indicating the F-URSP does not exhibit cytotoxicity.” (page 3, lines 75-79)

- p.3 l. 94; converted into a DC signal by the integrated rectifiers => generally electrical stimulation is biphasic; so dc has to be explained and contrasted.

Authors’ response: Thank you for pointing this out. Yes, biphasic pulses are usually used in clinic stimulations, while monophasic pulses (i.e., direct current) are often used in animal experiments for studying neuronal responses to the stimulations. The monophasic pulse is not preferred in clinical study because it could potentially cause skin burns or tissue damage when the current is high.^[1]

In our animal study, since we use a safe electric current and short stimulation pulse (< 1 ms, usually, deep brain stimulation stimulates for more than 10 mins^[1]). We are using monophasic stimulation because of its simplicity. The results of *ex vivo* retinal stimulation experiments show the feasibility of the ultrasound-induced scheme using our original design and preparation of F-URSP. Nevertheless, we would like to emphasize that there is no obstacle to generating biphasic stimulation by adding a simple passive electric circuit into our device for future translational study. We show here a classic electrical solution just for example here: 1. Two rectifiers with reversed poles are paralleled connected to the ultrasound piezo elements. They will generate two rectified DC signals with the same amplitude and duration. 2. A resistor-capacitor (RC) delay circuit is connected to one rectifier. Then it got a biphasic electrical signal. These will help us in optimizing the device in future translational studies.

(A classic electrical solution, just for example. A resistor-capacitor (RC) delay circuit is connected to one rectifier. Then it got a biphasic electrical signal.)

Following the reviewer's comment and suggestion, we have explained the DC signal and discussed more about the relevant optimization schemes in the **Discussion** section of the revised manuscript.

References:

[1] Fary, R. E. & Briffa, N. K. Monophasic electrical stimulation produces high rates of adverse skin reactions in healthy subjects. *Physiotherapy Theory and Practice* **27**, 246-251 (2011).

➤ **Line 104:** I think you mean to refer to Fig 1b, not 2b.

Authors' response: We thank the reviewer for the careful review. It was modified in the revised manuscript (line 104).

- p.4, l.124 designed and fabricated via a dicing-and-filling method to optimize the device performance (Fig. 2a) -> will you be presenting the methods or reference? Would be good to have dimensions on 2a.

Authors' response: Thanks for the question and suggestion. **Fig. 2a** was revised with a more detailed process and dimensions given. Additionally, we cited the relevant references^[1,2] in the revised manuscript (lines 128-129), and the detailed methods are shown in the **Methods** section.

References:

[1] Zhang, Y. Y. et al. Fabrication of PIMNT/Epoxy 1-3 composites and ultrasonic transducer for nondestructive evaluation. *IEEE Transactions on Ultrasonics, Ferroelectrics, and Frequency Control* **58**, 1774-1781 (2011).

[2] Wang, J. S. et al. Fabrication and high acoustic performance of high frequency needle ultrasound transducer with PMN-PT/Epoxy 1-3 piezoelectric composite prepared by dice and fill method. *Sensors and Actuators A-Physical* **318**, 112528 (2021).

- None of Fig 1 inset, 3 a, 5a give dimensions, particularly z-axis to give an idea of the size/scale of the array. In one picture, the 2D array dimensions, to depth/z-axis range for focusing are not evident.

Authors' response: We thank the reviewer for the comment. We have revised the mentioned figures (**Fig. 1 inset, Fig. 3a, and Fig. 5a**) with key components and aspect ratios labelled, especially the dimensions on the z -axis. For the 2D array, we have given the dimensions with depth/ z -axis range for focusing. Please see **Fig. 1a and Fig 5a** in the revised manuscript.

- One anticipated problem might be the curvature of the f-URSP; so not all pixel are at equidistance and therefore may not be focused enough or produce equivalent stimulation. But there are some serious limitations here.

Authors' response: We appreciate the reviewer's comment. Since the device would be attached to the eyeball with curvature, focusing on the different elements is indeed an issue worth considering.

First, the average radius of the normal adult eyeball is ~ 12 mm. Our device with an area of 8.5×8.5 mm² is attached to the anterior surface of the eyeball. According to the calculation, the outermost element and the center element have a parallel distance difference of ~ 0.74 mm from the transmitter, as shown in **Supplementary Figure 2c**. This distance difference can be completely compensated by modulating the depth of focus of the 2D transmitter.

Supplementary Figure 2 | c, Schematic diagram showing the size of the device on the eyeball.

Second, a 2D ultrasound array transducer is used as the ultrasound transmitter to emit a patterned acoustic field. Compared with a single probe in the focused or plane architecture, the multi-element 2D array transducer exhibits the scalability and programmability to allow steering and focusing at different depths of ultrasonic beamlines (**Supplementary Figure 17**) in the viewing direction to generate arbitrary patterns by modifying the transmitting amplitude and phase of each element. Therefore, the distance difference of the elements can be compensated by adjusting the direction and focus depth of the acoustic beams. These limitations can be solved in practical applications.

Following the reviewer's comments, the above discussions have been added into the revised manuscript (**Supplementary Figures 2 and 17**).

Supplementary Figure 17 | Simulated ultrasonic fields demonstrating ultrasound beam steering and focusing of a 2D array transducer. a, Simulated steering of ultrasonic beamlines. **b,** Simulated focusing at different depths of ultrasonic beamlines. The multi-element 2D array transducer exhibits the scalability and programmability to allow steering and focusing at different depths of ultrasonic beamlines in the viewing direction by modifying the transmitting amplitude and phase of each element.

- **Figure 1:** The schematic does not include ultrasound gel between the wearable transducers and flexible PCB. This is slightly misleading as to the system’s anticipated ultimate ease of use. Please add this to the figure and answer the question: what solutions will there be to the need for gel between the transducer and PCB?

Authors’ response: We are grateful that the reviewer point out this question. For effective ultrasonic transmission and coupling, degassed ultrasound gel (acoustic impedance ~ 1.5 MRayls) (EcoVue Ultrasound Gel, HR Pharmaceuticals, Inc.) was applied as the coupling medium between the 2D ultrasound array and the receiving piezo array through a custom frame for transmitting ultrasound to stimulate the elements. The ultrasound gel is confined between the 2D ultrasound array and the receiving piezo array by a custom frame. To better present the device operation process, the ultrasonic gel as a coupling agent is illustrated in the revised schematic diagram (**revised Fig. 1a**). In future translational applications, special flexible ultrasonic

coupling patches (a solid) would be customized with similar acoustic impedance and special configuration to closely fit the 2D ultrasound array (eyeglasses) with the receiving piezo array (eyeball surface) for ultrasonic transmission and fixation of the device architecture. In the revised manuscript, we illustrate it and give more discussion.

Fig. 1 | Design structure and working principle of the F-URSP. a, Schematic diagram showing the flexible ultrasonic device, with key components and aspect ratios labelled.

➤ Fig. 2 uses different terminology than Fig. 1

- The receiver array shows 32 pixels, but 4x4 (green -> are these 1x1 mm elements) and then even more MPN-PT rods; how many?

- 2h simulated potentials has no scale; I has 'normalized' scale for comparison, but value?

- Figure Two skips letters 'f' & 'g' (which may be intentional).

Authors' response: We appreciate the reviewer's questions.

(1) The receiver piezo-array has 32 pixels, and its specific arrangement is shown in **Supplementary Figure 2**. It is a 6×6 array with the four elements on the four corners removed.

For each element, its area is $1\text{ mm} \times 1\text{ mm}$ with a gap between two elements of 0.5 mm . Additionally, each element is composed of PMN-PT/epoxy 1-3 composite. The composite was prepared by dicing PMN-PT crystal and filling it with epoxy. The diced PMN-PT rods are uniformly distributed in the epoxy matrix. The length \times width \times height of each rod is $90\text{ }\mu\text{m} \times 90\text{ }\mu\text{m} \times 180\text{ }\mu\text{m}$, and the width of the kerfs is $30\text{ }\mu\text{m}$, as shown in **Fig. 2b**. Thus, there are ~ 64 PMN-PT rods in each piezo-element.

Supplementary Figure 2 | Schematic diagram showing the design structure of the F-URSP. a,b, Schematic diagram showing structure and dimensions of the device. The design refers to the size of the human eyeball for potential implant applications.

Fig. 2 | Structural optimization and simulation of the F-URSP. **a**, Schematic diagram showing the preparation process of the PMN-PT 1-3 composite for the harvesting array using a dicing-and-filling technology. **b**, Photographs of the diced PMN-PT 1-3 composite and the prepared harvesting piezo-elements. The composite consists of 90- μm -wide PMN-PT rods and 30- μm -wide kerfs. Each element possesses a 1 mm \times 1 mm element footprint with a spacing of 0.5 mm.

(2) We have added the color bar to **Fig. 2f**. The values were given to the color bars. Additionally, to facilitate the comparison of the ultrasound-induced potentials, we adjusted the potential ranges so that they are presented at the same color bar values.

(3) Thanks the reviewer. We have revised **Fig. 2** by using “f, g” instead of “h, i”.

Fig. 2 | Structural optimization and simulation of the F-URSP. f,g, Simulated piezo-potentials inside a bulk PMN-PT piezo-element (f) and a 1-3 composite element (g) induced by the same ultrasonic field (500 kPa).

➤ Fig. 3

- Can you justify, explain 3.3MHz frequency?

- The axial dimension to focusing is 2.5 cm; that seems high, but clarify that it is scalable, programmable

- Output voltage, its variability, scaling with different power, etc is clear. What's not clear is the stimulus threshold, and currents - it's the current that stimulates; so for the applied voltage, load determines the current: so, what is the expected load?

Authors' response: Thank the reviewer for the questions.

(1) In our work, the selection of ultrasound frequency of ~ 3.3 MHz is mainly based on the following considerations:

First, ultrasound technology has been applied for over 20 years and has an excellent safety record. The usual range of medical ultrasound for procedural guidance is 1-18 MHz^[1]. Therefore, for bio-implantable applications, the selection of ultrasonic frequency is preferred to be in this range, similar to other medical ultrasound devices.

Second, the wavelength of ultrasound directly affects its resolution (resolution $\propto \lambda$) because the focusing is wavelength-dependent. High-frequency ultrasound has shorter wavelengths for higher resolution. Our receiving piezoelectric array is with 32 elements, and each element size is $1 \times 1 \text{ mm}^2$. The gap between the two elements is 0.5 mm. To enhance the resolution of the acoustic beam without affecting the operation of the adjacent piezoelectric element, the wavelength of ultrasound should be less than 0.5 mm. For example, the wavelength 3.3-MHz ultrasound is ~ 0.47 mm, is possible to meet the resolution requirements.

Third, the attenuation of ultrasound energy is linearly dependent on the ultrasound frequency. For example, the ultrasound has a ~ 0.5-1 dB $\text{cm}^{-1} \text{ MHz}^{-1}$ acoustic attenuation coefficient in tissue^[2]. High frequency will cause high attenuation, thereby reducing transmission

efficiency. Therefore, combining the above aspects (the usual range of medical ultrasound, wavelength, and attenuation), 3.3-MHz ultrasound that combines high resolution with low attenuation but in the usual medical range was selected in this work.

Following the reviewer's suggestion, the above discussion on the frequency selection has been added to the revised manuscript (**Supplementary Note 3**).

(2) The axial dimension to focusing at 2.5 cm is for a single ultrasound transducer which is a choice of ultrasound source. Indeed, for a single ultrasound transducer, the depth of focus is generally unscalable and unchangeable once the preparation is complete. Here, we use this transducer mainly to discuss the frequency response (for ensuring that it is an ultrasound-induced output) and the efficiency of the output versus the input. Additionally, different array elements can be excited by moving it (fixed on a five-axis motor). However, this unscalable single probe is inefficient for stimulation. Therefore, we further used a 2D array as the excitation ultrasound source. Compared with a single probe in the focused or plane architecture, the multi-element 2D array transducer exhibits the ability to allow focusing and steering of ultrasonic beamlines (**Supplementary Figures 17-18**) in the viewing direction to generate arbitrary patterns by modifying the transmitting amplitude and phase of each element. Thus, it is scalable and programmable for a 2D array transducer.

(3) For retinal stimulation, the stimulus threshold is electrode voltage of ~ 0.8 V and pulse amplitude of ~ 125 μ A, which is referenced to commercially available Argus II visual prosthesis (Argus[®] II Retinal Prosthesis (Implant), Second Sight Medical Products, Inc, Sylmar, CA 91342, USA)^[3]. As a comparison, the voltage output of our ultrasound-induced device can reach to 4 V within the safety limit. If the device needs to meet this current threshold, the expected load should be less than ~ 30 k Ω . An average impedance modulus at the electrode-retina interface is few thousands of ohms (2-6 k Ω)^[3], thus resulting in that the current is sufficient to meet the current threshold. In addition, the output voltage can be flexibly adjusted by the input acoustic pressure.

References:

- [1] Bhatia, A. & Peng, P. In *Essentials of Pain Medicine*. 725-736 (Elsevier, 2018).

[2] Oglat, A. A. *et al.* Chemical items used for preparing tissue-mimicking material of wall-less flow phantom for doppler ultrasound imaging. *J. Med. Ultrasound* **26**, 123 (2018).

[3] Zhou, D. D., Dorn, J. D. & Greenberg, R. J. The Argus[®] II retinal prosthesis system: An overview. In Proc. *2013 IEEE International Conference on Multimedia and Expo Workshops (ICMEW)* (IEEE, 2013).

- Fig. 4 very nicely illustrates the beam focusing. Importantly, the graphs show spillovers to nearby pixels. The SNR is reasonably apparent; but the dB scale given does not give an idea whether this will lead to retinal pixel stimulation. i.e. there should be some clarity on above/below (or graded) threshold for phosphene/retinal excitation.

Authors' response: We appreciate the reviewer's comments.

As shown in the magnitude distribution in **Fig. 4**, spillovers to nearby pixels were indeed observed because the ultrasound beam is usually accompanied by sidelobes. But it can be seen from the magnitude distribution of the color bar that the magnitude of adjacent elements is almost lower than -10 dB, which means that the output power has been attenuated down to 30%. Therefore, we can control the amplitude of ultrasound input, ensuring the main focusing region can activate neurons while sidelobes are weaker than the threshold, to avoid out-of-focus activation. As previously reported in the literature, a low electrode voltage of ~ 0.8 V (Argus II Retinal Prosthesis (Implant) Second Sight Medical Products, Inc., Sylmar, CA 91342, USA) was usually applied to stimulate retinal cells. For the receiving array, its output voltage can be flexibly adjusted and proportional to the input. Combined with the measurements in **Fig. 3d**, thus, an input voltage of ~ 30 Vpp should be applied for the transmitter to achieve an output value higher than 0.8 V. Considering that the devices are applied to implants, as low power as possible under threshold conditions are met. In revised **Fig. 4d**, we have given the max voltage value (~ 2.1 V) and used dB scale to the maps of the receiving array elements because the output can be flexibly adjusted based on the input. For our device, an input voltage of ~ 30-50 Vpp is recommended for use, which can satisfy the effective activation of the central element (output voltage of central element ~ 0.99-2.1 V), and can also avoid spillovers to nearby pixels (output voltage of nearby element << 0.8 V).

Following the reviewer's comment, more discussion has been added to **Fig. 4** in the revised manuscript.

- Figure 5f-h: While the results are impressive, I do not agree on the clarity and reliability of the imaging results as claimed on lines 265-266. Please show multiple trials to demonstrate reliability in image reconstruction and provide quantitative measure of error between expected outcome and actual. For example, MSE per pixel between a simulated image reconstruction output and the results in 5f-h.

Authors' response: We appreciate the reviewer's comment and suggestion.

In the revised manuscript, we update the test results by optimizing the 2D array transducer and repeat the experiment several times to calculate the mean squared error (MSE) for each pixel, as shown in **Figure 5** and **Supplementary Figure 20**. The MSE ranges from 0.01 to 0.1. The error mainly comes from the uniformity of the emitted acoustic field. The 2D array transducer preparation process may lead to deviations in the elements, thus affecting the actual acoustic field emission. Additionally, spillovers to nearby pixels are observed because the ultrasound beam is usually accompanied by sidelobes. These can be further improved in the future by optimizing the preparation of the transducer.

Following the reviewer's suggestion, we updated the imaging results and revised the claims in the revised manuscript (page 12, lines 281-287).

Supplementary Figure 20 | Mean Squared Error (MSE) per pixel between a simulated image reconstruction output and the measured results. (a) Simulated image outputs. (b) Measured image outputs of multiple tests. (c) MSE per pixel between simulated outputs and the measured results. The MSE ranges from 0.01 to 0.1.

- Supplementary Fig. 6 is very useful regarding focusing field; but it does not map into the field of stimulation of retinal cells. This can be obtained by combining the electrical and acoustic field calculations (supplementary Note 1): what power is needed for comparable focusing, field of stimulation.

Authors' response: We appreciate the reviewer's comment. **Supplementary Figure 6** shows the simulated focusing field at the C plane to visualize the beam-focusing area and illustrate the beam's resolution. The simulated focusing field at the XY plane is shown in **Fig. 3b**. The results should be noted that the -6 dB (energy attenuated by half) lateral resolution of the emitted 3.3-MHz ultrasound beam is $\sim 390 \mu\text{m}$ near the focus point, which is below one element size (1 mm

$\times 1$ mm) of the receiving piezo-array, thus ensuring the less impact on adjacent elements when an element is excited. Receiving maps depending on one single element are shown in **Fig. 4d**.

As mentioned above, a low electrode voltage of ~ 0.8 V (Argus[®] II Retinal Prosthesis (Implant), Second Sight Medical Products, Inc., Sylmar, CA 91342, USA)^[1] was usually applied to stimulate retinal cells. For the receiving array, its output voltage can be flexibly adjusted and proportional to the input. Thus, an input voltage of ~ 30 Vpp should be applied for the transmitter to achieve an output value higher than 0.8 V. The acoustic intensity (I_{SPTA} : spatial peak temporal average intensity) corresponding to the focus area is ~ 0.229 W cm⁻². Nevertheless, there would be spillover/out-of-focus activation caused by the sidelobes in the ultrasound beams. Usually, the lateral distribution of the beam intensity involves the Gaussian distribution. The intensity attenuates extremely fast away from the central region. Considering that the devices are applied to implants, as low power as possible under threshold conditions are met. An input voltage of ~ 30 -50 Vpp is recommended for use. Due to the high lateral resolution, the central region still maintains a distribution of strong contrasts higher than the other regions, making it easy to achieve precise stimulation.

Following the reviewer' comment, we have re-edited the caption under **Supplementary Figure 6**, and the related discussion has been added to the Discussion section.

References:

[1] Zhou, D. D., Dorn, J. D. & Greenberg, R. J. The Argus[®] II retinal prosthesis system: An overview. In Proc. *2013 IEEE International Conference on Multimedia and Expo Workshops (ICMEW)* (IEEE, 2013).

REVIEWER COMMENTS

Reviewer #1 (Remarks to the Author):

The authors have suitably addressed the reviewers' comments and suggestion, including added data and further discussion / clarification.

Reviewer #2 (Remarks to the Author):

The authors have made extensive changes to address the prior critique, which much improved the paper. Some points still need to be addressed:

Line 258: 'The pixels on the acoustic path can generate a higher potential (~ 2.1 V) that is above the voltage threshold of ~ 0.8 V for retinal excitation, while the other pixels, which deviate from the sound beam, possess a high magnitude attenuation (< -10.8 dB) with voltage well below 0.8 V.'

I don't understand 'high magnitude attenuation'. I assume that pixels outside the focal spot receive less acoustic excitation, which is not 'attenuation.'

Fig 6: The ex vivo retina was stimulated by a 'V' shaped acoustic pattern. The authors state 'The correspondingly observed fluorescence-enhanced region on the retina matches the expected excitation position (a "V" pattern).' I think this statement requires some stretch of faith, as Fig 6c shows perhaps a 'U' pattern. It would be more convincing if Figs 6b and 6c are presented in the same way format (6b in 2D, not 3D) and on the same scale. Also, the legend states that 6b is a simulation. Why weren't actual measurements made?

The authors mention heating of the natural crystalline lens of the eye by ultrasound, but they also should consider that the lens will both attenuate and refract (defocus) acoustic beams generated at the cornea. Probably lens extraction (as performed routinely in cataract surgery) would be advantageous to bypass this issue.

The authors explain that degassed ultrasound gel will be used to couple the eyeglass-array to the corneal surface-mounted array. This would work in principle, but would be difficult in practice, as the gel would tend to slosh around and fall out with patient motion or even standing up. Perhaps a solid gel like an Aquaflex gel pad might work, but even this seems difficult.

The authors focus on design of the the surface-mounted transmit array and the retinal stimulating array. However, the eyeglass array aspect of the device is not well thought-out. They now mention future integration of a camera to stimulate the eyeglass array, but this is not shown in Fig 1. The eyeglass-mounted array as shown in the figure has no way to convert light to an organized image to be projected acoustically onto the corneal surface-mounted array. The figure should offer a plausible means for addressing this aspect of the device. The authors should modify the figure to include a camera and processor or at least state in the figure legend that a camera and chip will be designed in the future to excite the array.

Reviewer #3 (Remarks to the Author):

This is a remarkably strong paper and the technology and its demonstration is impressive. The authors should be commended for their extremely diligent response (and pardon me for taking the time to wade through the long response and the revision). There are many answers in the rebuttal and they are quite satisfactory. Many design parameters are described, as also the extensive evaluations. The ex vivo and in vivo in transgenic mouse now raise even more confidence in the feasibility. The authors should be commended for adding the calcium imaging experiments with an ex vivo retina dissected from Thy1-GCaMP6f transgenic mice The safety aspects are good, temperature to cell studies. A number of other design issues, focusing, resolution/speed etc, 3.3 MHz, etc. are addressed. Data in comparison with Argus II look promising.

The remainder weakness, rebutted by the authors, is the testing on cell lines for safety in lieu of the actual target cells or the tissue. Frankly, it defocuses the paper. At this stage the paper is more technology and an early-stage demonstration of the evidence of neural/retinal stimulation. Safety profile for now is reasonable (PC-3cells, 3 days) although nowhere near adequate for making a compelling case for in vivo, chronic. This may be a bar too high to hold up publication of an exciting work.

Hence, if you still insist on putting the cell-line safety work from the paper, then move most of it (e.g. Fig. 6) to the supplement. Keep the focus on in vivo, retinal cell demonstration (new, refined experiments and data are indeed adequate).

Minor issues

The rebuttal is very informative. Some excerpts should be succinctly added,

- E.g. polyene-C/coating, protection from PZT etc
- The authors reply on focusing, temporal persistence, frequency of stimulation; in lieu of substantial data, these numbers or observations are/can be in Discussion. Of course, any supplementary data will strengthen the paper.
- Could the scaling of the transmitter-receiver field mismatch, focusing, and how that actually helps (lower receiver field) be better illustrated? Many figures/dimensions are in the supplements, but selectively put in the main figures as these will be read, and scale etc are important to note.
- The answer on biphasic is inaccurate. The R-C solution is, I am sorry to say, naive. The whole idea of biphasic is to achieve charge balance and R-C filtering does not do that. It may be better to simply admit what's here and how you will go to biphasic in the future. Fig. 1 is beautiful, but not scaled or rigorous. Dimensions are in the supplement, but not the scaling to the eye.

REVIEWER COMMENTS

Reviewer #1 (Remarks to the Author):

The authors have suitably addressed the reviewers' comments and suggestion, including added data and further discussion / clarification.

Authors' response: We appreciate the reviewer for the valuable comments.

Reviewer #2 (Remarks to the Author):

The authors have made extensive changes to address the prior critique, which much improved the paper. Some points still need to be addressed:

Authors' response: We appreciate the reviewer's constructive comments to improve our paper. We present the point-by-point response for each comment as follows.

Line 258: 'The pixels on the acoustic path can generate a higher potential (~ 2.1 V) that is above the voltage threshold of ~ 0.8 V for retinal excitation, while the other pixels, which deviate from the sound beam, possess a high magnitude attenuation (< -10.8 dB) with voltage well below 0.8 V.'

I don't understand 'high magnitude attenuation'. I assume that pixels outside the focal spot receive less acoustic excitation, which is not 'attenuation.'

Authors' response: We thank the reviewer for pointing this out, and it is an excellent comment. It is true that pixels outside the focal spot receive less acoustic excitation and thus produce a smaller voltage. The "attenuation" used here is not very appropriate. Therefore, following the reviewer's suggestion, we have revised this in the revised manuscript (lines 256-259), copied below:

"The pixels on the acoustic path can generate a higher potential (~ 2.1 V) above the voltage threshold of ~ 0.8 V for retinal excitation⁴³, while the other pixels outside the focal spot receive less acoustic excitation and, therefore, produce lower electrical outputs with voltages below 0.8 V."

Fig 6: The ex vivo retina was stimulated by a 'V' shaped acoustic pattern. The authors state 'The correspondingly observed fluorescence-enhanced region on the retina matches the expected excitation position (a "V" pattern).' I think this statement requires some stretch of faith, as Fig 6c shows perhaps a 'U' pattern. It would be more convincing if Figs 6b and 6c are presented in the

same way format (6b in 2D, not 3D) and on the same scale. Also, the legend states that 6b is a simulation. Why weren't actual measurements made?

Authors' response: We appreciate the reviewer's professional comment and suggestion. We have measured the acoustic pressure distribution emitted by a 2D array transducer through a hydrophone probe, as shown in the revised **Fig. 6b** in the revised manuscript. It is a "V"-shaped pattern. The measured acoustic pressure distribution and the observed fluorescence-enhanced region are presented in the same format (2D) and on the same scale. The pattern region, combined with the measured acoustic pressure distribution and observed fluorescence-enhanced region, is better matched to a "V"-shaped pattern, albeit with a little distortion. This is mainly limited by the number of array elements, their size, and the resolution of the current device. We have given more discussion in the **Discussion section** to enhance the pattern reconstruction capability of the device through using a high-frequency 2D array with smaller and denser piezo-units, and larger channel counts, etc., in future work.

Fig. 6 | Living retina stimulation of F-URSP. b. Measured acoustic pressure distribution of a "V"-shaped pattern emitted by a 2D array transducer through a hydrophone probe. The color bar indicates the normalized acoustic pressure. The maximum value is 0.45 MPa. c. Calcium imaging of the living retina at stimulated levels. d. Time course of calcium transients during stimulation.

The authors mention heating of the natural crystalline lens of the eye by ultrasound, but they also should consider that the lens will both attenuate and refract (defocus) acoustic beams generated at the cornea. Probably lens extraction (as performed routinely in cataract surgery) would be advantageous to bypass this issue.

Authors' response: We are very grateful to the reviewer for the comments and suggestions on this part. We agree that the lens will both attenuate and refract (defocus) acoustic beams generated at the cornea. Therefore, following the reviewer's suggestion, lens extraction (as performed routinely in cataract surgery) was discussed and added to the "**Discussion**" section (lines 410-412) to facilitate the future application of this ultrasound device toward clinical translation, copied below:

"In addition, lens extraction (as performed routinely in cataract surgery) at the time of device implantation would be more advantageous considering that the lens will both attenuate and refract (defocus) the acoustic beams."

The authors explain that degassed ultrasound gel will be used to couple the eyeglass-array to the corneal surface-mounted array. This would work in principle, but would be difficult in practice, as the gel would tend to slosh around and fall out with patient motion or even standing up. Perhaps a solid gel like an Aquaflex gel pad might work, but even this seems difficult.

Authors' response: We appreciate the reviewer's professional comment and suggestion. Ultrasonic glue can work in principle, but it will encounter difficulties in practical application. Following the reviewer's suggestion, a solid gel like an Aquaflex gel pad (an aqueous, flexible ultrasound standoff) would be a better solution to prevent sloshing around and falling out with patient motion. In future research toward translational applications, we will adopt a scheme based on the reviewer's suggestion to optimize the device structure and promote its practicality.

Therefore, in the revised manuscript, more discussion on the above optimization options has been added to the "**Discussion**" section (lines 406-408) to facilitate future translational studies of the ultrasound device, copied below:

"A solid gel like an Aquaflex gel pad (an aqueous, flexible US scaffold) would be customized in a unique configuration to closely match the 2D array (eyeglasses) and the receiving piezo-array (eyeball surface) to prevent sloshing around and falling out with patient motion, thus enhancing its practicality."

The authors focus on design of the the surface-mounted transmit array and the retinal stimulating array. However, the eyeglass array aspect of the device is not well thought-out. They now mention future integration of a camera to stimulate the eyeglass array, but this is not shown in Fig 1. The eyeglass-mounted array as shown in the figure has no way to convert light to an organized image to be projected acoustically onto the corneal surface-mounted array. The figure should offer a plausible means for addressing this aspect of the device. The authors should modify the figure to include a camera and processor or at least state in the figure legend that a camera and chip will be designed in the future to excite the array.

Authors' response: Thanks to the author for the valuable suggestions. To illustrate the glasses that can convert light to an organized image to be projected acoustically onto the corneal surface-mounted array, a micro-camera for videography and an integrated chip for optical picture-to-program processing are integrated into the glasses, as shown in the revised schematic **Fig. 1a**. In addition, the figure legend was revised to state that a micro-camera for videography and an integrated chip for optical picture-to-program processing will be designed and integrated into the glasses in the future to excite the array.

Fig. 1 | Design structure and working principle of the F-URSP. a, Schematic diagram showing the flexible ultrasonic device, with key components and aspect ratios labelled. The implantable F-URSP consists of a high-performance piezo-composite array, rectifiers, and an electrode array

integrated into a flexible PCB. Each piezo-channel (CH1-CH32) in the array works individually and can convert the transmitted US wave into electricity which is rectified and then used for electrical stimulation of retinal neurons through the integrated electrodes. The correspondingly induced action potentials would be conducted to the central visual pathway via the optic nerve to produce visual perception. Therefore, the F-URSP architecture leverages the advantage of the programmable US beam to realize the visual projection of complex patterns wirelessly. A micro-camera for videography and an integrated chip (IC) for optical picture-to-program processing will be designed and integrated into the glasses in the future to excite the array. *P*, Parylene-C/coating (10 μm), protection for PZT; *ML*, matching layer (35 μm), enhancement of acoustic transmittance; *PC*, piezo-composite (180 μm), acoustic-electric conversion element; *PCB*, printed circuit board (150 μm), flexible circuit carrier; *R*, rectifier (200 μm), AC-DC conversion; *AE*, Au/Cr electrode (100 nm), electrodes of the piezo-element.

Reviewer #3 (Remarks to the Author):

This is a remarkably strong paper and the technology and its demonstration is impressive. The authors should be commended for their extremely diligent response (and pardon me for taking the time to wade through the long response and the revision). There are many answers in the rebuttal and they are quite satisfactory. Many design parameters are described, as also the extensive evaluations. The ex vivo and in vivo in transgenic mouse now raise even more confidence in the feasibility. The authors should be commended for adding the calcium imaging experiments with an ex vivo retina dissected from Thy1-GCaMP6f transgenic mice. The safety aspects are good, temperature to cell studies. A number of other design issues, focusing, resolution/speed etc, 3.3 MHz, etc. are addressed. Data in comparison with Argus II look promising.

Authors' response: We thank the reviewer for his/her positive comments on our work and the constructive comments to enhance our work. We respond to each comment point by point as follows.

The remainder weakness, rebutted by the authors, is the testing on cell lines for safety in lieu of the actual target cells or the tissue. Frankly, it defocuses the paper. At this stage the paper is more technology and an early-stage demonstration of the evidence of neural/retinal stimulation. Safety profile for now is reasonable (PC-3 cells, 3 days) although nowhere near adequate for making a compelling case for in vivo, chronic. This may be a bar too high to hold up publication of an exciting work.

Hence, if you still insist on putting the cell-line safety work from the paper, then move most of it (e.g. Fig. 6) to the supplement. Keep the focus on *in vivo*, retinal cell demonstration (new, refined experiments and data are indeed adequate).

Authors' response: We appreciate the reviewer's professional comment and suggestion. Indeed, this paper is more technology and an early-stage demonstration of the evidence of neural/retinal stimulation. Therefore, in the revised manuscript, we have moved the current data generated with the PC-3 cell line to the Supplementary Information and kept the focus on retinal cell

demonstration, as shown in **Fig. 6** and **Supplementary Figure 22**. In the future, we will continue in-depth research in this direction with histopathology for *in vivo* experiments.

Minor issues

The rebuttal is very informative. Some excerpts should be succinctly added,

- E.g. paylene-C/coating, protection from PZT etc.

Authors' response: We thank the reviewer for the suggestion. In the revised manuscript, we have added some excerpts in the text, as shown in the highlighted revision in the legend of **Fig. 1**.

- The authors reply on focusing, temporal persistence, frequency of stimulation; in lieu of substantial data, these numbers or observations are/can be in Discussion. Of course, any supplementary data will strengthen the paper.

Authors' response: Thanks to the reviewer for the suggestion. For the replies on focus, temporal persistence, and frequency of stimulation, we have added them in the **Discussion** section in the revised manuscript (lines 365-378), copied below:

“A US 2D array was used to dynamically generate different acoustic focusing patterns to excite the receiving pixels for electrical stimulation of retinal neurons. The area, distance, and intensity of the acoustic focusing can be flexibly adjusted through controlling the US input parameters to ensure that the main lobe can activate neurons, but sidelobes are weaker than the threshold to avoid out-of-focus activation. As a consequence, the pixels under US excitation produce sufficient electrical outputs (e.g., voltage ~ 0-4.3 V, power density ~ 0-22.7 mW cm⁻²) that are higher than the stimulus parameters (e.g., an electrode voltage ~ 0.8 V) used by an EM-based commercially available visual prostheses⁴³. US-induced electrical stimulation elicits the same neuronal response as conventional electrical stimulation methods. For every 0.3 milliseconds of stimulation, the neuronal spike lasts about 10 milliseconds; for the evoked calcium signal, it can last up to 30 seconds. In addition, the 2D array can dynamically generate different patterns in a frame rate higher than 1 kHz. Conventional electrical stimulation studies found that neuron responses have their own limitations. For example, Argus II, the existing commercial retina prosthesis, only works at 10 Hz frame rate while its device supports a frame rate higher than 120

Hz¹¹. Therefore, the stimulation frequency of our ultrasonic visual prosthesis can be adjusted according to the actual demand.

- Could the scaling of the transmitter-receiver field mismatch, focusing, and how that actually helps (lower receiver field) be better illustrated? Many figures/dimensions are in the supplements, but selectively put in the main figures as these will be read, and scale etc are important to note.

Authors' response: We appreciate the reviewer's question and suggestion. To better illustrate the scaling of the transmitter-receiver field mismatch, focusing, etc., we have modified the relevant schematic figures, such as **Supplementary Figure 2**. The figures containing dimensions were moved from the **Supplementary Information** to the **main figures** for easier reading, as shown in **Fig. 1b**.

Fig. 1 | Design structure and working principle of the F-URSP. a, Schematic diagram showing the flexible ultrasonic device, with key components and aspect ratios labelled. The implantable F-

URSP consists of a high-performance piezo-composite array, rectifiers, and an electrode array integrated into a flexible PCB. Each piezo-channel (CH1-CH32) in the array works individually and can convert the transmitted US wave into electricity which is rectified and then used for electrical stimulation of retinal neurons through the integrated electrodes. The correspondingly induced action potentials would be conducted to the central visual pathway via the optic nerve to produce visual perception. Therefore, the F-URSP architecture leverages the advantage of the programmable US beam to realize the visual projection of complex patterns wirelessly. A micro-camera for videography and an integrated chip (IC) for optical picture-to-program processing will be designed and integrated into the glasses in the future to excite the array. *P*, Parylene-C/coating (10 μm), protection for PZT; *ML*, matching layer (35 μm), enhancement of acoustic transmittance; *PC*, piezo-composite (180 μm), acoustic-electric conversion element; *PCB*, printed circuit board (150 μm), flexible circuit carrier; *R*, rectifier (200 μm), AC-DC conversion; *AE*, Au/Cr electrode (100 nm), electrodes of the piezo-element. **b. Schematic diagram showing structure and dimensions of the F-URSP. The design refers to the size of the human eyeball (radius ~ 12 mm) for future implant applications.**

- The answer on biphasic is inaccurate. The R-C solution is, I am sorry to say, naive. The whole idea of biphasic is to achieve charge balance and R-C filtering does not do that. It may be better to simply admit what's here and how you will go to biphasic in the future. Fig. 1 is beautiful, but not scaled or rigorous. Dimensions are in the supplement, but not the scaling to the eye.

Authors' response: We thank the reviewer for his/her comment and suggestion on this R-C solution. We admit that R-C filtering is indeed a simple model, and it is difficult to implement the whole idea of biphasic to achieve charge balance. In the future, to achieve biphasic stimulus pulses, a custom application-specific integrated circuit (ASIC) consisting of power management, control, and stimulation modules for modulating the ultrasound-induced power and stimulation (biphasic) will be designed into the device. For example, Jacob T. Robinson et al. reported a wireless miniature device combined with an ASIC to safely stimulate peripheral nerves^{R1}, as shown in **Fig. R1** below. We will refer to their solution to achieve biphasic stimulus pulses in the future.

Fig. R1. ASIC providing digitally programmable stimulation^{R1}. The ASIC is used to wirelessly receive power and data, and consists of power management, data recovery, control, and stimulation modules to drive programmable stimulation.

Following the reviewer’s suggestion, ASIC-based solution in the future has been added to the “**Discussion**” section (lines 412-415) in revised manuscript, copied below:

“Fourth, the biphasic stimulation solution would be adopted by incorporating a custom application-specific integrated circuit (ASIC) consisting of power management, control, and stimulation modules into the ultrasonic device for translational research, rather than monophasic stimulation that cannot achieve charge balance⁶³.”

For the schematic **Fig. 1**, we have modified it in the revised manuscript as much as possible according to the scale of the eye. Thanks again to the reviewer for this suggestion.

References:

R1. Chen, J. C., Kan, P., Yu, Z. et al. A wireless millimetric magnetoelectric implant for the endovascular stimulation of peripheral nerves. *Nat. Biomed. Eng.* (2022). <https://doi.org/10.1038/s41551-022-00873-7>.

REVIEWERS' COMMENTS

Reviewer #2 (Remarks to the Author):

The authors have responded to concerns and comments in my critique.

This paper demonstrates an important avenue towards vision restoration in patients with retinopathies, specifically, no-functional photoreceptors.

While some concerns remain regarding practical implementation, these are outweighed by the significance of acoustic stimulation of the retina.

The paper is in my judgement now suitable for publication.